# Compute-Optimal LLMs Provably Generalize Better with Scale

**Marc Finzi**
Carnegie Mellon University

**Sanyam Kapoor**
New York University

**Diego Granziol**
PureStrength AI

**Anming Gu**
Boston University

**Christopher De Sa**[*]
Cornell University

**J. Zico Kolter**[*]
Carnegie Mellon University

**Andrew Gordon Wilson**[*]
New York University

## Abstract

Why do larger language models generalize better? To explore this question, we develop generalization bounds on the pretraining objective of large language models (LLMs) in the compute-optimal regime, as described by the Chinchilla scaling laws. We introduce a novel, fully empirical Freedman-type martingale concentration inequality that tightens existing bounds by accounting for the variance of the loss function. This generalization bound can be decomposed into three interpretable components: the number of parameters per token, the loss variance, and the quantization error at a fixed bitrate. As language models are scaled up, the number of parameters per data point remains constant; however, both the loss variance and the quantization error decrease, implying that larger models should have *smaller* generalization gaps. We examine why larger models tend to be more quantizable from an information theoretic perspective, showing that the rate at which they can integrate new information grows slower than their capacity on the compute-optimal frontier. From these findings we produce a scaling law for the generalization gap, showing that our bounds become stronger in a predictable way.

## 1 Introduction

Large language models (LLMs) have demonstrated a remarkable general purpose problem solving capacity across a wide range of complex tasks, from classical NLU (Brown, 2020), forecasting (Gruver et al., 2023), mathematics (Trinh et al., 2024), spatial reasoning (Patel & Pavlick, 2022), and many other areas. For a large majority of individual tasks, model capabilities increase monotonically as the next token prediction loss from the pretraining objective decreases.

A conceptually useful story about the learning process involves the model accommodating predictive subprograms of progressively larger computational depth and complexity. During pretraining, shallow details like word frequencies, syntax, and grammar are absorbed first, followed by higher level structures such as facts, relations, and idioms, eventually giving way to yet higher level patterns. For reasons not yet well understood, this process manifests in the pretraining objective as a power law for LLMs and other generative models on natural data. The frontier of best achievable performance given a fixed computational budget $C$ obeys a predictable power law relationship $L(C) \propto C^{-\gamma}$ over many orders of magnitude (Kaplan et al., 2020), varying considerably with the kind of data (Henighan et al., 2020) but only weakly with model architecture and training method (Bahri et al., 2021).

Effort in quantifying *what* this relationship is in a given domain and *how* it varies as model size and dataset size are traded off has been extremely valuable in guiding where resources are spent in constructing more capable AI models (Brown, 2020; Besiroglu et al., 2024; OpenAI, 2023; Dubey et al., 2024) and charting a path for the future. In this work, we target the *why* of scaling laws. While mathematically simple toy models or toy data are valuable, we aim to study the *why* of scaling laws on real models and real data by focusing on one contribution to the scaling law curve: the token-wise generalization gap. Constructing a generalization bound sensitive enough to capture the small differences between architectures and yet simple enough to write down in a short formula is

---

[*]Equal advising.

likely impossible; however, even the broad strokes of behavior such as how generalization scales with compute have not been addressed. Thus, here we focus high level understanding rather than algorithmic intervention.

We can observe that in order for the generalization gap not to eventually dominate the contributions to the test loss (which has not yet been observed), it must be that the generalization gap decreases at least as fast as the approximation gap does. Deeply understanding the success of the pretraining scaling of LLMs paradigm requires being able to predict that this would be the case.

In order to construct the relevant generalization bounds, we introduce a novel empirical Freedman concentration inequality (Freedman, 1975). Our generalization bound highlights three critical components—the ratio of parameters per token in compute-optimal scaling (which is roughly constant), the token-wise loss variance (which decreases with model size), and the performance gap between quantized and unquantized models (which also decreases with model size). As an alternative to quantization, we bound the information transfer between dataset and the model, showing that the information content in the model grows sublinearly with model size, and thus the complexity decreases with model size. These components collectively contribute to a predictable reduction in the generalization gap as models grow larger.

## 2 BACKGROUND

### 2.1 GENERALIZATION BOUNDS

At a high level, we are interested in the expected test error (population risk) $\mathbb{E}_{X' \sim p_{\mathcal{D}}}[R_{h(X)}(X')]$ for a given model (hypothesis) $h$ depending on the training set $X$ but evaluated on a test set $X'$ sampled from the data distribution $p_{\mathcal{D}}$. One conceptually convenient way of breaking down this quantity is into the irreducible error, approximation gap, and generalization gap:[1]

$$\mathbb{E}_{X' \sim p_D}[R_{h(X)}(X')] = \underbrace{R_*(X)}_{\text{Irreducible Error } E} + \underbrace{R_{h(X)}(X) - R_*(X)}_{\text{Approximation Gap } A} + \underbrace{\mathbb{E}_{X' \sim p_D}[R_{h(X)}(X')] - R_{h(X)}(X)}_{\text{Generalization Gap } G}.$$

The first term describes the entropy of natural text, e.g. the amount of truly random information content in the data, which cannot be further explained even when knowing the true data generating process. The second term describes the approximation gap, capturing the extent to which the trained model is able to fit the training data. This term combines both model capacity, e.g. as described by universal approximation theorems (Cybenko, 1989), as well as optimization via how well the training algorithm is able to find the given solution. Finally, we have the generalization gap, capturing the extent to which training and testing performance diverge on account of overfitting to the statistically irrelevant regularities in $X$. Though generalization bounds focus on the last term, all three quantities are of interest for understanding LLM behavior. Empirically, it has been observed that the generalization gap for LLMs (at least in the low epoch regime) tends to be extremely small compared to the other two terms and we aim to make sense of why this is the case.

Among the simplest generalization bounds is the finite hypothesis with prior generalization bound applied to IID data (Shalev-Shwartz & Ben-David, 2014). With probability at least $1 - \delta$,

$$\mathbb{E}_{X' \sim p_{\mathcal{D}}}[R_{h(X)}(X')] - R_{h(X)}(X) \leq \Delta \sqrt{\frac{\log 1/P(h) + \log 1/\delta}{2m}}$$

where $m$ is the number of IID data points, $\Delta$ is an upper bound on the range of values the risk can take, and $P(h)$ is a prior distribution over hypotheses in a discrete hypothesis class $\mathcal{H}$. With a judicious choice of prior, $\log 1/P(h)$ can be related to the compressed size of the model measured in nats (Lotfi et al., 2022).

During text pretraining, the individual tokens are not sampled IID. Thus, a generalization bound requires treating entire documents (often thousands of tokens) as the elements the empirical risk is computed over. Note that modern language models have hundreds of times more parameters than

---

[1]We note this differs from the commonly referred to estimation-approximation error breakdown (Bottou & Bousquet, 2007) or the bias-variance decomposition (Brown & Ali, 2024); however, the train error-generalization gap is more useful for our purposes.

documents they were trained on. With the help of very extreme compression methods and using smoothing to bound $\Delta$, it is possible to construct nonvacuous bounds (Lotfi et al., 2024a). However, the required compression (greater than 100 times) is so severe that it cripples model performance.

In a recent work, Lotfi et al. (2024c) explore breaking down generalization into tokenwise generalization, e.g. how the loss varies with each individual predicted token being resampled under the distribution but keeping the context the same. Splitting up the training dataset $X$ into the sequence of tokens $[X_k]_{k=1}^D$, the authors bound

$$T = \frac{1}{D} \sum_{k=1}^D \mathbb{E}[R_h(X_k \mid X_{<k}) \mid X_{<k}] - R_h(X),$$

where $R_h(X_k \mid X_{<k})$ is the negative log likelihood for token $k$ given the context $X_{<k}$, and the expectation is taken with respect to $p(X_k|X_{<k})$ from the data distribution. The authors bound $T$ using Azuma's inequality to arrive at a bound scaling as $\Delta\sqrt{\frac{\log 1/P(h)}{2D}}$. Using a novel empirical Freedman type inequality, we bound the same quantity $T$ but improve upon this bound, reducing the leading term to $\Delta\frac{\log 1/P(h)}{D}$.

## 2.2 CHINCHILLA SCALING LAWS

A key insight from the current machine learning paradigm is that the dataset should not be considered a fixed quantity. Rather than optimizing to find the best model for a given dataset, one should instead try to find the best performing model and dataset for a given computational budget. Hoffmann et al. (2022) describes the optimal allocation of resources for increasing the size of the model and increasing the size of the dataset under the assumption that data is plentiful relative to the computational budget.

Let $N$ be the number of parameters and $D$ be the number of training tokens. In the one epoch regime of LLM pretraining, the negative log likelihood (NLL) loss is well-approximated by the power law

$$R(N, D) = E + \frac{A}{N^\alpha} + \frac{B}{D^\beta},$$

where $A, B$ are empirically estimated constants, exponents $\alpha, \beta$ have similar values, and $E$ is the irreducible error. Optimizing $N(C)$ and $D(C)$ under the constraint of a fixed compute budget $C \approx 6ND$ (Kaplan et al., 2020), one arrives at

$$N^*(C) = G(C/6)^a, \qquad D^*(C) = G^{-1}(C/6)^b$$

for constants $G = \left(\frac{\alpha A}{\beta B}\right)^{1/(\alpha+\beta)}$, $a = \beta/(\alpha + \beta)$, and $b = \alpha/(\alpha + \beta)$, as derived in Hoffmann et al. (2022).

Within the margin of statistical error, we have $a = b = 0.5$ in the optimal allocation of compute (Besiroglu et al., 2024). Therefore, the ratio of parameters per token, $N^*(C)/D^*(C) = G^2$, is a fixed constant. Evaluating the constants from Besiroglu et al. (2024), we have $G^2 \approx 1/20$. We remark that many open source models optimize performance amortized over both training time and inference time compute (Sardana et al., 2023), which leads to "smaller than Chinchilla optimal models," e.g. models with a ratio $N/D < G^2$, and similarly when repeating data in the data constrained setting Muennighoff et al. (2024). In the context of this paper, we will assume the Chinchilla optimal scaling $N/D = G^2$, and remark that any generalization bounds we construct would only be tighter if the ratio $N/D$ is smaller.

To test our theory, we use the open source Pythia model family (Biderman et al., 2023) ranging from 70 million to 12 billion parameters. Unlike other open source LLMs, we have full access to both the Pythia model checkpoints from training and the Pile dataset they were trained on (Gao et al., 2020), which is required for our analysis. From these intermediate

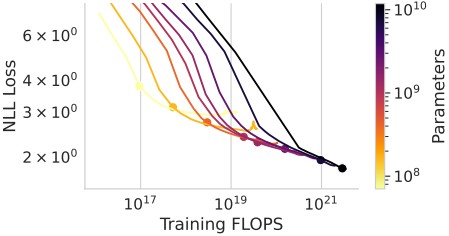

Figure 1: Pythia models and checkpoints chosen along the compute-optimal frontier (checkpoints given by the marked values).

checkpoints, we choose the set of models along the compute-optimal frontier to match $N/D = G^2 \approx 1/20$, reflecting the choice for number of training steps and model size that one would have made optimizing only for performance at the given computational budget. The chosen checkpoints are plotted in the training frontier of these models in Figure 1.

## 3 GENERALIZATION BOUND

In this section, we build the components used in constructing our final generalization bound stated in Theorem 3.4. To capture the relevant behavior, we derive a new concentration inequality for martingales. We apply a prior weighted union bound to this concentration inequality so that we can apply it to models in a large hypothesis class, taking advantage of the low complexity inherent in compressible models. Bounding the worst case loss behavior using prediction smoothing, we apply this bound to LLMs.

At a high level, we can motivate the overall behavior of the bounds as follows. On account of the compute optimal scaling, as LLMs are scaled, the ratio of parameters to training tokens remains a fixed constant, $G^2 \approx 1/20$, less than one. However as the models improve, the per token standard deviation of the loss decreases, and at a predictable rate of approximately $c + 1/\sqrt{N}$. If the per token variation is small, then a sufficiently sensitive concentration inequality will translate this into a tighter concentration around the mean. Simultaneously, the amount of information stored per parameter, and thus the compressed size of the model given a suitable compression scheme, appears to decrease with scale at the compute optimal frontier. Combining these observations, we produce generalization bounds showing that the gap between train and test shrinks as these models are scaled up.

### 3.1 AN EMPIRICAL FREEDMAN'S CONCENTRATION INEQUALITY

**Theorem 3.1.** *Let $(X_k)_{k=1}^n = X_1, \ldots, X_n$ and $(Y_k)_{k=1}^n$ be sequences of random variables adapted to the filtration $(\mathcal{F}_k)_{k=0}^n$ where $X_k$ is $\mathcal{F}_k$ measurable and $Y_k$ is $\mathcal{F}_{k-1}$ measurable. Assume the difference between the two is bounded below: $A_k = (Y_k - X_k)/\Delta > -1$ for some $\Delta \geq 0$. Let $K$ be any finite subset of $(0, 1)$. Then, with probability at least $1 - \delta$ over the probability space (filtered by $(\mathcal{F}_k)_{k=0}^n$),*

$$\frac{1}{n} \sum_{k=1}^n \left( \mathbb{E}[X_k \mid \mathcal{F}_{k-1}] - X_k \right) \leq \Delta C + \Sigma \sqrt{C}, \tag{1}$$

*where $C := \frac{1}{n} \log \frac{|K|}{\delta}$, and*

$$\Sigma = \Sigma(C, \Delta, \{Y_k - X_k\}_{k=1}^n, K) := \min_{s \in K} \Delta \sqrt{C}(1 - s)/s + \frac{\Delta}{\sqrt{C}} \frac{1}{n} \sum_{k=1}^n v(s A_k)/s$$

*and $v(x) = x - \log(1 + x)$.*

The proof is provided in Appendix A.1. This concentration inequality states that the sequence of random variables concentrates on their conditional means with a term $\Sigma$ depending on the empirical variation of the loss value. We note that $\Sigma$ can be viewed as a variance term. As we show in Appendix A, using a small $K$, the variance proxy can be upper bounded: $\Sigma \leq 2\sqrt{\frac{1}{n} \sum_{k=1}^n (X_k - Y_k)^2}$, explicitly related to the empirical variance but with the mean replaced by $Y_k$. Although the minimization form above is unwieldy, it produces significantly tighter estimates of $\Sigma$ (a factor of $\sim 5x$ smaller). When the loss variation $\Sigma$ is small, concentration to the mean happens at a rate linear in the complexity $C$ rather the slower $\sqrt{C}$ rate. The finite set $K$ serves merely as a device to control how finely $s$ is optimized, and can be set for example as a uniformly spaced grid in $(0, 1)$.

The concentration inequality we present here provides the core result for our generalization bounds, and to the best of our knowledge it is the first martingale concentration inequality to incorporate a variance term which can be evaluated on the original data. We can view this bound as aiming to achieve the benefits that Freedman's inequality has over Azuma's inequality while being fully empirical, replacing the population variance with a fully empirical proxy. Our approach is analogous to the fully empirical Bernstein bound derived in Maurer & Pontil (2009), but in the martingale rather than IID setting. Unfortunately, the proof technique of Maurer & Pontil (2009) does not carry over to

the martingale case and instead we take a different approach. We derive our concentration inequality in Theorem A.5 making use of a proxy $Y_k$ that is $\mathcal{F}_{k-1}$-measurable but which can take the place of $\mathbb{E}[X_k \mid \mathcal{F}_{k-1}]$ in the variance. In practice, we choose this quantity to be the mean of the model NLL under resampling of the given token according to the *model* distribution in place of the data distribution.

## 3.2 EXTENDING TO A DISCRETE HYPOTHESIS CLASS

From the concentration inequality in equation 1, we derive the following discrete hypothesis class generalization bound.

**Lemma 3.2.** *Let $X_1, \ldots, X_n$ be a sequence of (possibly dependent) random variables. Let $R_h(X_k \mid X_{<k})$ denote the risk for element $X_k$ given the previous elements of the sequence for hypothesis $h$ in a countable hypothesis class $\mathcal{H}$. Let $p_h(X_k \mid X_{<k})$ be any (hypothesis dependent) distribution over $X_k$ conditioned on $X_{<k}$. Consider a prefix free coding of each $h \in \mathcal{H}$ and let $L(h)$ be the length of that code measured in nats. Let $K$ be a finite subset of $\mathbb{R}^+$. Assuming $R_h(X_k \mid X_{<k}) - \mathbb{E}_{Y_k \sim p_h}[R_h(Y_k \mid X_{<k})] \leq \Delta$ for some $\Delta > 0$, we have that simultaneously for all $h \in \mathcal{H}$, with probability at least $1 - \delta$,*

$$\frac{1}{n} \sum_k \mathbb{E}[R_h(X_k \mid X_{<k}) \mid X_{<k}] \leq \frac{1}{n} \sum_k R_h(X_k \mid X_{<k})) + \Delta \mathcal{C} + \Sigma \sqrt{\mathcal{C}}, \qquad (2)$$

*where the complexity $\mathcal{C}$ is given by*

$$\mathcal{C} := \frac{L(h) + \log |K|/\delta}{n}$$

*and $\Sigma = \Sigma(\mathcal{C}, \Delta, \{A_k\}_{k=1}^n, K)$ is as in Theorem 3.1 for $A_k = R_h(X_k \mid X_{<k}) - \mathbb{E}_{Y_k \sim p_h}[R_h(Y_k \mid X_{<k})]$.*

We provide the proof in Appendix A.2.

## 3.3 WORST CASE BEHAVIOR AND SMOOTHING

The last component of our bounds is the smoothing to bound the worst case behavior of the model, which in general for the negative log likelihood can be arbitrarily large. We employ the prediction smoothing idea from Lotfi et al. (2024a), where the model is mixed with a uniform distribution over the tokens with a given mixing parameter. Unlike application in previous work, we optimize over this parameter analytically so that we can remove it from the bounds and evaluation entirely, instead of merely as a tool for constructing bounds while all evaluations are with the unsmoothed model.

**Lemma 3.3.** *For the categorical negative log likelihood objective $\hat{R}_h = -\frac{1}{n} \sum_{k=1}^n \log p_h(X_k \mid X_{<k})$ on $V$ classes and $C \in \mathbb{R}^+$, there exists a prediction smoothed model $p_s(\cdot) = (1-\alpha)p_h(\cdot) + \alpha/V$ which has a worst case loss $\Delta_s = \sup_{X_k, X_{<k}} -\log p_s(X_k \mid X_{<k}) \leq \log(V/\alpha)$, and the risk satisfies*

$$\hat{R}_s + C\Delta_s \leq \hat{R}_h + C \log V + \sqrt{2C}, \qquad (3)$$

*for some value $\alpha \in (0, 1)$ (approximately $C/(1 + C)$).*

We provide the proof in Appendix A.3.

## 3.4 GENERALIZATION BOUND FOR COMPUTE OPTIMAL LANGUAGE MODELS

Finally, we assemble these three components into a generalization bound that we can empirically evaluate for language models. Combining the prediction smoothing bound with Theorem 3.2 applied to the smoothed quantized model produces our main result. Note that each term in the expression has an interpretable meaning.

**Theorem 3.4.** *Let $X_1, \ldots, X_D$ be the sequence of $D$ (possibly dependent) tokens formed from concatenating each sequence in the dataset together into a single stream of tokens. Let $R_h(X_k \mid X_{<k}) = -\log p_h(X_k \mid X_{<k})$ denote the NLL for element $X_k$ given the previous elements for a given model $h$ and with vocabulary size $V$ and $N$ parameters. Let $\hat{R}_h = \frac{1}{D} \sum_{k=1}^D R_h(X_k \mid X_{<k})$*

be the empirical risk and $R_h = \frac{1}{D}\sum_{k=1}^{D}\mathbb{E}[R_h(X_k \mid X_{<k}) \mid X_{<k}]$ be the tokenwise expected risk for that model. Let $K$ be a finite subset of $(0,1)$. For any given quantization $q$ of $h$ using $b$ bits per parameter with expected risk $R_q$, there exists a label smoothed and quantized model $sq$ with $R_{sq}(X_k \mid X_{<k}) = (1-\alpha)R_q(X_k \mid X_{<k}) + \alpha/V$ for fixed $\alpha \in (0,1)$ which, with probability $1-\delta$, achieves a tokenwise population risk

$$R_{sq} \leq \hat{R}_h + \underbrace{\mathcal{C}\log V}_{Random\ Guess\ NLL} + \underbrace{\Sigma\sqrt{\mathcal{C}}}_{Loss\ Variation} + \underbrace{\sqrt{2\mathcal{C}}}_{Smoothing\ Cost} + \underbrace{(\hat{R}_q - \hat{R}_h)}_{Quantization\ Gap}, \tag{4}$$

where the complexity $\mathcal{C}$ is given by

$$\mathcal{C} = \left(\tfrac{N}{D}\right)b\log 2 + \tfrac{1}{D}\log\tfrac{|K|}{\delta},$$

and $\Sigma = \Sigma(\mathcal{C}, \Delta, \{A_k\}_{k=1}^{n}, K)$ (defined in Theorem 3.2) can be upper bounded in terms of the empirical loss variance:

$$\Sigma \leq 2\sqrt{\frac{1}{D}\sum_{k=1}^{D}\left(R_q(X_k \mid X_{<k}) - \mathbb{E}_{Y_k \sim p_h}[R_q(Y_k \mid X_{<k})]\right)^2}.$$

To make sense of the bound, let's consider the various terms. The bounded quantity on the left hand side, $R_{sq}$, is the expected tokenwise risk of the smoothed and quantized version of hypothesis $h$. The bound is written in terms of the empirical risk of the original model $h$, with $\hat{R}_q - \hat{R}_{sq}$ controlled by the smoothing cost and quantization gap. Typically, the largest contribution to the bound is $\mathcal{C}\log V$, e.g. the complexity times the negative log likelihood of random guessing. The loss variation relates to how spread the empirical loss is and can be seen as a model realizability term. If there existed a $0$ loss model in the model class, then this term could be brought to $0$; however, given the nonzero entropy of natural text, this will not be the case. Note that as models improve and approach the irreducible error, so too will the empirical loss variation.

In this setup, the complexity $\mathcal{C}$ is just the ratio of parameters to data points, $\frac{N}{D} = G^2$, times the number of bits per parameter used in the quantization $b$, plus a negligible additional term. The decreased complexity of using fewer bits for $b$ trades off with the quantization gap, and in principle this parameter should be optimized to achieve the best bound. As all terms in the expression can be evaluated empirically, we can determine how much of the empirical observation it can explain and how much remains to be understood.

To get a sense for the scale of the different terms, consider the following typical scenario in LLMs. The averaged negative log likelihood loss is measured in nats per token, where a nat is $\log_2 e$ bits. For simplicity, suppose we have vocabulary size $V = 50000$ (so $\log V \approx 11$), quantization $b = 3$, and $G^2 = 1/20$, which yields $\mathcal{C} \approx 1/9$. $\Sigma$ varies with model scale but is of scale $1/10$, and the quantization gap is around $1/10$. Evaluating these terms, we see that $R_{sq} - \hat{R}_h \leq 11/9 + 1/30 + 1.4/3 + 1/10 \approx 1.8$ nats per token, and we see that the random guess and smoothing terms contribute most to the size of the bound. For perspective, the empirical risk $\hat{R}_h$ itself is around $2$ nats per token and the boundary between vacuous and nonvacuous bounds is at $\log V \approx 11$ nats per token.

## 4 EMPIRICAL EVALUATION

As Theorem 3.4 is fully empirical, we simply need to empirically evaluate the loss variation term $\Sigma$ along with the quantization gap and we can evaluate the generalization bound. We compute these quantities on the given Pythia checkpoints on the Pile dataset on which they were trained and quantized using GPTQ (Frantar et al., 2023) to $b = 4$ bits per parameter, and we evaluate the bounds with failure probability $\delta = 0.01$. The results are shown in Figure 2 for the Chinchilla compute-optimal checkpoints within the Pythia model family.

We compute $\Sigma$ with $K$ given by 1000 equally spaced points between $[0, 1]$, excluding the endpoints. We estimate the risk and loss variation on an IID subsample from the collection of token-context pairs in the training dataset of size $10^4$ and bound the difference from the full training set evaluation and the $10^4$ sized subsample with a simple Hoeffding bound. We note that largest 12B parameter model failed to quantize properly (possibly due to the learning rate drop as it was the only checkpoint taken at the

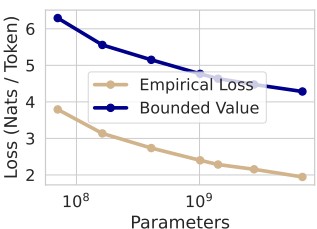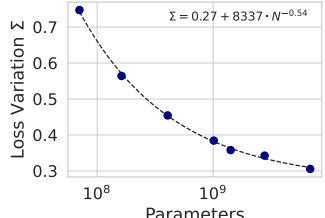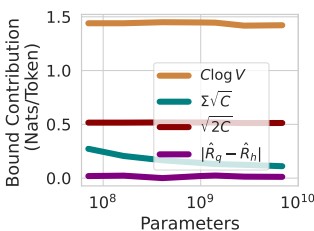

Figure 2: **Left:** A direct comparison of our evaluated generalization bound, and the empirical loss as a function of model scale. As the model is scaled up, our bound improves just like the empirical loss. **Center:** Loss variation $\Sigma$ entering into the generalization bound. As the loss deviation decreases, so does the largest term in our bound. **Right:** Comparison of the relative scale of the contributions to Theorem 3.4. Here we use a fixed 4 bit quantization of the parameters.

end of training) and so we removed it from the evaluation. We must pick out the compute-optimal checkpoints from training runs that were not designed for this purpose. For the two smallest models (70m and 160m parameters) where the compute optimal training duration is early into the training run and thus sparsely sampled with checkpoints, and the closest models consistently have too small a value of $N/D$, biasing these two initial data points towards lower values. For these two points, we compute the relevant quantities $D, \Sigma, \hat{R}_h, \hat{R}_q, \Sigma$ by linearly interpolating using the parameter $N/D$ with the target value of $G^2$.

In Figure 2, we can observe several points. In Figure 2 (center), we see that the loss variation decreases with model size as $0.27 + 8337N^{-0.54}$, approximately $1/\sqrt{N}$ with a constant offset. As additional compute is spent in model training, predictions narrow and the loss variation decreases. Like with the irreducible error $E$, it converges to a minimum value presumably related to the varentropy of the distribution. In Figure 2 (right), we break down the individual contributions to the generalization bound. The quantization error and loss variance contributing a fraction of a nat per token and decrease with scale, while the $\mathcal{C} \log V$ term and smoothing terms contribute the majority to the bounds and do not decrease with scale. Figure 2 (left) shows the comparison with the bound value $R_{sq}$ and the empirical risk $\hat{R}_h$. The fact that the quantization gap at a fixed number of bits decreases with model size has been observed to a much greater extent in other work (Chee et al., 2024; Tseng et al., 2024) with more advanced quantization methods and with fewer bits per parameter. This property suggests that if $b$ were able to be freely optimized in the bound, the complexity $\mathcal{C}$ would actually decrease with model size, and we explore evidence for and consequences of this idea in the following section.

## 5 COMPRESSIBILITY AND THE SUBLINEAR INFORMATION GROWTH IN LLMS

While not obvious from efficient quantization algorithms like GPTQ (Frantar et al., 2023), there are good reasons to believe that the model complexity term $L(h)/D$ decreases with model scale on the compute-optimal frontier.

### 5.1 QUANTIZABILITY FROM THE HESSIAN SPECTRUM AND QUIP

So far we have split the compressed size of the model $L(h)$ featured in the complexity term into the number of parameters $N$ times the number of bits per parameter used in the quantization $b$: $L(h) \leq bN \log 2$. In this splitting, increased compressibility of a model shows up in terms of requiring a smaller number of bits $b$ to achieve a given quantization gap $\hat{R}_q - \hat{R}_h$. In Appendix B, we provide a theoretical argument using the QuIP quantization framework (Chee et al., 2024) for why we should expect that larger models can be more easily quantized. If the Hessian around the solution weights is positive semi-definite (PSD) and the spectrum decays sufficiently rapidly, then we should expect the quantization error to decrease with model size. In Section B.3, we investigate the Hessian spectrum empirically finding that it indeed decays sufficiently quickly (though not always PSD). Unfortunately, the version of QuIP needed to construct this argument cannot be run in practice due to the impractically large computational constraints. Empirically it has been observed that practical

quantization algorithms also reveal that larger models are more quantizable (Tseng et al., 2024), though the effect is not very pronounced with the GPTQ algorithm we use here.

Alternatively, we present a more abstract information-theoretic argument to provide evidence for the fact that $L(h)/D$ decreases with model scale even if we do not have an explicit compression scheme to achieve it.

## 5.2 INFORMATION ACCUMULATION IN LLMS

At initialization, the information content in a large neural network is extremely small, requiring only the model architecture and a random seed to be fully specified. As training progresses, information transfers incrementally from the dataset to the model weights. This transfer can be quantified using prequential coding (Rissanen, 1984; Dawid, 1984) and algorithmic information theory.

Let $K(X)$ be the (prefix) Kolmogorov complexity of the dataset $X$ (the length of a shortest self-delimiting program producing $X$). From the symmetry of information property, $K(X, h) = K(h) + K(X|h) + c$, where $c$ is a small constant. Rearranging, $K(h) = K(X, h) - K(X|h) - c$ measures the information content in $h$ as the difference between the size of the smallest program that codes $X$ and $h$ jointly and the smallest program that codes $X$ given $h$. As described in Blier & Ollivier (2018); Voita & Titov (2020) and more specifically in Zhang et al. (2020), one can use prequential coding to provide an estimate for an upper bound on $K(X, h) - K(X|h)$.

A prequential code (Rissanen, 1984; Dawid, 1984) provides a means to code the tuple of data and probability model $(X, h)$ by using codes derived from intermediate snapshots of the model as it processes and updates on each successive data point in $X$ during training. As setup, one considers the sender and receiver each to have an identical copy of the model at initialization $h_0$ (along with the randomness seed if randomness is used in the training algorithm). From this model at initialization, the initialized probability model $h_0$ can be used to encode data in its domain with using arithmetic coding (or any entropy stream code) using $-\log_2 p_{h_0}(X_1)$ bits, which can be decoded by the receiver using $h_0$. With the first data point $X_1$ transmitted, both the sender and receiver train on this data point yielding identical models $h_1$. From here the process can be repeated coding the subsequent data point with coded using $h_1$ and so forth until the entire dataset has been transmitted. Using an entropy code for the transmission, the entire code for the transmission need not be greater than $-\sum_{k=1}^{D} \log_2 p_{h_{k-1}}(X_k|X_{<k})$ bits, the area under the loss curve. With this transmission the receiver can reconstruct the entire dataset $X = (X_1, X_2, \ldots X_D)$ and the sequence of models produced during training $[h_1, h_2, \ldots, h_D]$. While this prequential code is not a prefix free code, it can be converted into one with logarithmic extra bits (see Section A.2), or merely a small constant extra bits if the vocabulary size $V$ and dataset size $D$ are prespecified, which we will assume from now on. With the prequential code, $K(X, h)$ can be upper bounded for a generative model $h$ in terms of the area under the loss curve, and notably this is true regardless of the number of parameters in $h$ or the number of bits needed in a more direct coding scheme. Instead, the information in the model is determined by the information in the data.

While not a strict lower bound, $K(X|h)$ can be estimated from entropy coding of the data using the generative model as suggested in Zhang et al. (2020), and with this strategy the codelength for $X$ given $h$ is $-\sum_{k=1}^{D} \log_2 p_{h_D}(X_k|X_{<k})$, the loss for the final model. Assembling the two together, up to small constant factors,

$$K(h)\log(2) \leq \sum_{k=1}^{D} \left[ R_{h_{k-1}}(X_k|X_{<k}) - R_{h_D}(X_k|X_{<k}) \right]. \tag{5}$$

We evaluate this expression numerically for the Pythia models, however we can also gain some insight by examining the asymptotic scaling of this quantity with the size of the dataset. For a reasonable approximation for the loss along the training trajectory, one can use the Chinchilla scaling law $R(N, D) = E + AN^{-\alpha} + BD^{-\beta}$. Noting that $\frac{1}{D}\sum_{k=1}^{D} f(k) \to \frac{1}{D}\int_1^D f(x)dx$ as $D \to \infty$,

$$K(h)\log(2) \leq \left( \sum_{k=1}^{D} R(N, k) \right) - DR(N, D) \to \frac{\beta}{1-\beta} D^{1-\beta} = O(D^{1-\beta}). \tag{6}$$

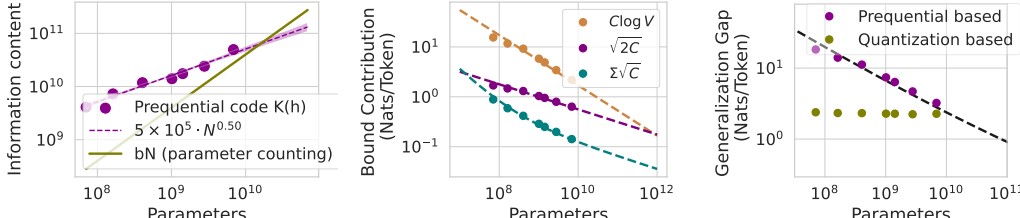

Figure 3: **Left:** Information content contained in the model as upper bounded by $K(h)$ from the information transfer prequential coding approach vs parameter counting and quantization. Fitting a power law to the prequential $K(h)$ yields $6 \times 10^5 \cdot N^{0.5 \pm 0.1}$. While parameter counting gives a better upper bound over the range of Pythia models, the sublinear scaling of the prequential bound means that it overtakes it eventually, somewhere around 30B sized models. **Center:** The contributions of the various terms to our generalization bounds when using prequential coding complexity, along with their power law fits. **Right:** Comparison of generalization bounds produced by the prequential vs quantization based approaches. While the prequential bounds are worse, they follow a power law and improve substantially with scale.

From this we obtain an insightful result: the information content in the model grows sublinearly with training dataset size, with a coefficient depending on the scaling law.

As shown in Figure 3 (left), using the empirical loss curves to evaluate $\sum_{k=1}^{D} \left[ R_{h_{k-1}}(X_k|X_{<k}) - R_h(X_k|X_{<k}) \right]$ and fitting the results to a power law, we get $6 \times 10^5 \cdot N^{0.5 \pm 0.1} \propto D^{0.5}$. In contrast, from the asymptotics using the scaling law to approximate the loss $\beta = 0.37$ evaluating (Besiroglu et al., 2024) yields $D^{1-\alpha} = D^{0.63}$ which is not far off. While the empirical values for the upper bound lie above the straightforward value one gets from quantization and parameter counting $bN$ over the range of Pythia models, the curves predict a crossover point at $\approx 20B$ parameter models.

### 5.3 IMPLICATIONS FOR GENERALIZATION BOUNDS

If we apply this observation to upper bound the complexity featured in our generalization bounds (Theorem 3.4) $\mathcal{C} = \frac{L(h) + \log |K|/\delta}{D} = K(h) \log(2)/D + \frac{\log |K|/\delta}{D} = O(D^{-\beta})$, we see that the complexity will actually decrease with the size of the dataset even as the ratio with parameters is held constant. With this scaling, we can derive a version of the generalization bound Theorem 3.4 without needing to consider quantization or considering the number of explicit parameters in the model, provided that the Chinchilla scaling law holds.

We evaluate the non asymptotic generalization bound of Theorem 3.4 but using the complexity derived from empirical prequential coding bound in Figure 3 (right) and break it down into the scaling of the individual terms (center), with the $\Sigma$ term scaling law extrapolated from the fit in Figure 2. Like before, the $\mathcal{C} \log V$ term dominates; however, the $\sqrt{2\mathcal{C}}$ smoothing term threatens to overtake it with very large model sizes. We can see that the bounds based on the prequential coding are worse than their quantization counterparts; however, the bounds improve with scale and can be extrapolated with scaling laws. Considering only the asymptotics, the generalization gap of Theorem 3.4 will be dominated by the scaling of the smoothing term $\sqrt{2\mathcal{C}}$: $R_s - \hat{R}_h = O(D^{-\beta/2})$. To speculate, it seems likely that with a more sophisticated approach for dealing with the unbounded loss, the $\sqrt{\mathcal{C}} = O(D^{-\beta/2})$ could be removed, letting the $O(D^{-\beta})$ shine through. If that were the case, then the tokenwise generalization gap could indeed be hidden within the $D^{-\beta}$ of the original scaling law.

## 6 ADDITIONAL RELATED WORK

**Generalization Bounds.** Historically, generalization bounds for neural networks have been limited by their large parameter count, though significant progress has been made in explaining generalization behavior (Dziugaite & Roy, 2017; Zhou et al., 2018; Arora et al., 2018; Lotfi et al., 2022), with PAC-Bayes providing a convenient unifying framework (Catoni, 2007). Lotfi et al. (2024a) constructed

the first non-vacuous generalization bounds for LLMs using prediction smoothing and extreme compression with subspace LoRA (Hu et al., 2021).

While Lotfi et al. (2024a) focused on document-level bounds, Lotfi et al. (2024b) used Azuma's inequality for token-level martingale-based bounds. We adopt this approach but improve the complexity from $O(\sqrt{\mathcal{C}})$ to $O(\mathcal{C})$ through loss variation. Related work has constrained context learning in LLMs (Li et al., 2023) and explored generalization in vision-language models (Akinwande et al., 2023).

Chugg et al. (2023) developed generalization bounds for both IID and martingale settings generalizing many previous results; however, these bounds are not fully empirical and thus can't be applied in the setting we require. Closest to our work, Maurer & Pontil (2009) derived a fully empirical Bernstein inequality, their technique doesn't extend to non-IID martingale settings.

**Post-Training Quantization.** For hardware efficiency, significant research has explored reducing model precision post-training while preserving performance (Hassibi et al., 1993; Hubara et al., 2021; Yao et al., 2022; Dettmers et al., 2022). Empirically, 3-4 bits provide a reasonable compression-performance tradeoff, with recent work pushing to 1.58 bits per parameter (Ma et al., 2024) and even binary networks showing promise (Liu et al., 2024).

For this work, we use GPTQ (Frantar et al., 2023) with 4-bit quantization, which achieves efficient extreme quantization through iterative weight column rounding. Alternative approaches include QuIP (Chee et al., 2024), which uses *incoherence* in approximate Hessian estimation to suppress outliers, and its improved variant QuIP# (Tseng et al., 2024). We leverage the analysis from these two papers in Appendix B to shed some light on how the quantizability scales with model size due to the spectrum of the Hessian.

## 7    DISCUSSION

We have provided generalization theory to better explain why large language models trained in the compute-optimal regime generalize, with particular attention on how generalization changes with scale. For the term that contributes the most to the generalization bound, we are able to improve the scaling over previous work from $\sqrt{\mathcal{C}} \log V$ to $\mathcal{C} \log V$, while remaining fully empirical. We explore two approaches for constraining model complexity in the generalization bounds, directly via quantization and parameter counting, and indirectly, via information transfer as quantified by prequential coding. While the quantization approach yields tighter bounds at Pythia model scale, the information transfer approach reveals that information in the model grows at a rate that is sublinear in the size of the dataset, and consequently the generalization gap must also decrease with scale.

While we believe that these insights help advance understanding, there are a number of limitations of our approach and many questions that remain unaddressed. As previously mentioned, the $\sqrt{2\mathcal{C}}$ smoothing term seems pessimistic and could likely be improved with a different approach. Additionally, while the information transfer argument provides evidence that the complexity of a model is low based on the training curve, it falls short of explaining *why* the complexity is low. In principle, the training curve could look different if it did not follow the Chinchilla power law scaling, leading to a different information transfer rate. Similarly, the Hessian based argument explains why larger models are more quantizable given the scaling of spectrum of the Hessian, but why the Hessian spectrum has this empirical behavior remains unexplained. Furthermore, it seems likely that the $1/\sqrt{N}$ in the loss variation term could be explained theoretically.

Even more broadly, our generalization bounds constrain only the token-wise generalization gap. While it is intuitive that generalizing well on next token prediction over the training contexts should imply generalization on the full sequences, we are not aware of work that does so, and this gap remains to be understood. Similarly, constraining generalization on the NLL objective over data drawn from the natural distribution may be less pertinent. Instead, it may be more relevant to constrain the gap between the quality metrics of model generations and natural data. Further removed, there is the question of why the training loss scales in the way that it does, and how does that relate to approximation theory and the architecture of the model? Though many questions remain, we hope that the techniques here can yield generalizable insights for tackling this broader set of problems.

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

## A PROOFS

### A.1 A FULLY EMPIRICAL MARTINGALE FREEDMAN CONCENTRATION INEQUALITY.

In this section, unless otherwise specified, $\log$ is used to denote the natural logarithm. We start with three technical lemmas.

**Lemma A.1.** *Consider the function $v(a) = a - \log(1 + a)$. Let $\mu \in \mathbb{R}$. For any random variable $Z$ with $\mathbb{E}[Z] = 0$ that satisfies $Z - \mu > -1$, we have*

$$\mathbb{E}[\exp(Z - v(Z - \mu))] = (1 - \mu)e^{\mu} \leq 1.$$

*Proof.* We see that

$$\mathbb{E}[\exp(Z - v(Z - \mu))] = e^{\mu}\mathbb{E}[\exp(Z - \mu - v(Z - \mu))] = e^{\mu}\mathbb{E}[1 + (Z - \mu)].$$

As $\mathbb{E}[Z] = 0$, we know $\mathbb{E}[\exp(Z - v(Z - \mu))] = (1 - \mu)e^{\mu}$. Additionally as $1 - \mu \leq e^{-\mu}$, we have $\mathbb{E}[\exp(Z - v(Z - \mu))] \leq 1$, as desired. $\square$

**Lemma A.2.** *Consider the function $v(a) = a - \log(1 + a)$. For all $a \in (-1, \infty)$, we have*

$$v(a) \leq a^2/(1 + a)$$

*Proof.* Let $f(a) = a^2/(1 + a) - v(a)$. By direct calculation, we see that $f'(a) = a/(1 + a^2)$, which is a strictly negative function passing through 0 at $a = 0$. As $f(0) = 0$, we have $f(a) \geq 0$ for all $a \geq 0$. Note that $\lim_{a \to -1^-} f(a) = +\infty$, so $f(a)$ must be positive on $(-1, 0)$. The claim follows. $\square$

For the following result, consider the filtered probability space $(\Omega, \mathcal{F}, \{\mathcal{F}_k\}_{k \in \mathbb{N}}, \mathbb{P})$, and we consider expectations with respect to $\mathbb{P}$.

**Lemma A.3.** *Let $X_1, \ldots, X_n$ be a sequence of $\mathcal{F}_k$-measurable random variables. Let $Y_1, \ldots Y_n$ be any other sequence of $\mathcal{F}_{k-1}$-measurable random variables such that the difference is bounded above: $A_k = Y_k - X_k > -\Delta$ for some $\Delta \geq 0$. Define $C = \frac{1}{n} \log \frac{1}{\epsilon}$, and let $B = \frac{1}{n} \sum_k \left(\mathbb{E}[X_k \mid \mathcal{F}_{k-1}] - X_k\right)$. For any $0 < t < 1/\Delta$ and simultaneously for all $n$, we have*

$$\mathbb{P}\left[tB \leq C + \frac{1}{n}\sum_{k=1}^{n} v(tA_k)\right] \geq 1 - \epsilon. \tag{7}$$

*Proof.* Let $v(a) = a - \log(1 + a)$. Define the random variable

$$M_k = \exp\left(t(\mathbb{E}[X_k \mid \mathcal{F}_{k-1}] - X_k) - v(t(Y_k - X_k))\right).$$

Consider $Z = t(\mathbb{E}[X_k \mid \mathcal{F}_{k-1}] - X_k)$ and $\mu = t(\mathbb{E}[X_k \mid \mathcal{F}_{k-1}] - Y_k)$. By construction, we have that $\mathbb{E}[Z \mid \mathcal{F}_{k-1}] = 0$ and $Z - \mu = t(Y_k - X_k) > -t\Delta > -1$. Thus, applying Lemma A.1, we have

$$\mathbb{E}[M_k \mid \mathcal{F}_{k-1}] \leq 1.$$

Therefore $U_n = \prod_{k=1}^{n} M_k$ is a supermartingale. By Ville's inequality (Ville, 1939), we have

$$\sup_n U_n \leq \frac{\mathbb{E}[U_0]}{\epsilon} \leq 1/\epsilon$$

with probability at least $1 - \epsilon$. When this holds, using the definition of $U$ and taking the log of both sides, we have for all $n$,

$$t\sum_{k=1}^{n}\left(\mathbb{E}[X_k \mid \mathcal{F}_{k-1}] - X_k)\right) - \sum_{k=1}^{n} v(t(Y_k - X_k)) \leq \log\frac{1}{\epsilon}. \tag{8}$$

Defining $B = \frac{1}{n}\sum_k \left(\mathbb{E}[X_k \mid \mathcal{F}_{k-1}] - X_k\right)$, $C = \frac{1}{n}\log\frac{1}{\epsilon}$, and $A_k = Y_k - X_k$, by rearrange equation 8, we obtain

$$tB \leq C + \frac{1}{n}\sum_{k=1}^{n} v(tA_k), \tag{9}$$

as desired. $\square$

**Corollary A.4.** *Let $X_1, \ldots, X_n$ be a sequence of $\mathcal{F}_k$-measurable random variables. Let $Y_1, \ldots Y_n$ be any other sequence of $\mathcal{F}_{k-1}$-measurable random variables such that the difference is bounded below: $A_k = (Y_k - X_k)/\Delta > -1$ for some $\Delta \geq 0$. Let $K$ be a finite subset of $(0,1)$. Then, with probability at least $1 - \delta$,*

$$\frac{1}{n} \sum_{k=1}^{n} \left( \mathbb{E}[X_k \mid \mathcal{F}_{k-1}] - X_k \right) \leq \Delta C + \Sigma \sqrt{C}, \tag{10}$$

*where $C := \frac{1}{n} \log |K|/\delta$, and*

$$\Sigma(C, \Delta, \{X_k - Y_k\}_{k=1}^{n}, K) := \min_{s \in K} \Delta \sqrt{C}(1 - s)/s + \frac{\Delta}{\sqrt{C}} \frac{1}{n} \sum_{k=1}^{n} v(sA_k)/s$$

*Proof.* Let $s = t\Delta$. Apply a union bound to Theorem A.3 over the different values of $s$ in $K$, and take the one that minimizes the bound. Rearrange and isolate terms yields the desired result. □

**Theorem A.5.** *Let $X_1, \ldots, X_n$ be a sequence of $\mathcal{F}_{i-1}$-measurable random variables. Let $Y_0, \ldots Y_{n-1}$ be any other sequence of $\mathcal{F}_{i-1}$ measurable sequence of random variables such that the difference is bounded above: $X_k - Y_k \leq \Delta$ for some $\Delta \geq 0$. Define $V = \frac{1}{n} \sum_k (X_k - Y_k)^2$ and let $\delta \in (0, 1)$. Then, with probability at least $1 - \delta$, we have*

$$\frac{1}{n} \sum_k \left( \mathbb{E}[X_k \mid \mathcal{F}_{k-1}] - X_k \right) \leq \Delta \mathcal{C} + 2\sqrt{V\mathcal{C}}, \tag{11}$$

*where $\mathcal{C} \leq n^{-1} \left( \log 1/\delta + 4 \log \log n/\delta + 6 \right)$.*

*Proof.* Starting from Lemma A.3, we apply Lemma A.2 of $v(a) \leq a^2(1 + a)$. For our convenience, here we will define $A_k = Y_k - X_k$. We have

$$tB \leq C + \frac{1}{n} \sum_k \frac{t^2 A_k^2}{1 + tA_k}$$

$$\leq C + \frac{1}{n} \sum_k \frac{t^2 A_k^2}{1 - t\Delta}, \tag{12}$$

where the second inequality follows from the assumption that $A_k \geq -\Delta$.

Finally, by defining a variance term $V = \frac{1}{n} \sum_k A_k^2 = \frac{1}{n} \sum_k (X_k - Y_k)^2$ and rearranging equation 12, we see

$$0 \leq t^2(V + B\Delta) - (B + \Delta C)t + C, \tag{13}$$

which we recall holds with probability at least $1 - \epsilon$.

**Inequality Sketch:** This inequality is very close to what we need. The approach would be to optimize over $t$ and then read off the constraint on $B$. The minimizer of the quadratic is at $t^* = \frac{B + \Delta C}{2(V + \Delta B)}$. Plugging in this value, one would arrive at

$$(1/4) \frac{(B + \Delta C)^2}{(V + \Delta B)} - (1/2) \frac{(B + \Delta C)^2}{(V + \Delta B)} + C \geq 0$$

Rearranging,

$$\frac{1}{4}(B + \Delta C)^2 - B\Delta C \leq VC$$

$$\frac{1}{4}(B - \Delta C)^2 \leq VC$$

and finally,

$$B \leq \Delta C + 2\sqrt{VC}.$$

At a high level, this determines the overall form of Theorem A.5; however, some technical complications arise from the fact that $t$ must be deterministic and chosen ahead of time, e.g. it must not

depend on the random variables $B$ and $C$. Instead we will consider optimizing $t$ over a discrete set of possibilities (not depending on $B$ or $C$), and consider a union bound over the different possibilities.

**Full derivation:** Consider the quadratic, equation 13. Its minimizer is given by

$$t^* = \frac{B + \Delta C}{2(V + \Delta B)}.$$

Consider two cases: $t^* \geq \frac{1}{\Delta}$ and $t^* < \frac{1}{\Delta}$. If $t^* \geq \frac{1}{\Delta}$, then rearranging and solving for $B$, we see

$$B \leq \Delta C - 2V/\Delta,$$

which is strictly less than the value $\Delta C + 2\sqrt{VC}$, and we are done.

Therefore it suffices to consider the case $t^* < 1/\Delta$, where we can apply Lemma A.3. Note that this result applies for a single $t$, so it cannot be directly applied to $t^*$. Instead, we will turn to quantization and apply a union bound. Note that if $B > 0$, using that $V \leq \Delta^2$, we have $t^* \geq \frac{\Delta C}{2\Delta^2} = \frac{C}{2\Delta}$. Therefore we only need to consider the range: $t^* \in (\frac{C}{2\Delta}, \frac{1}{\Delta}) =: T$.

Drawing inspiration from floating point numbers, consider a discrete set $Q$ defined as

$$Q = \left\{ \frac{1}{\Delta} 2^{-b} \left( 1 + \frac{k}{K} \right) \;\middle|\; k = 0, 1, ..., K - 1, b \in \mathbb{N}^+ \right\}$$

for some $K \in \mathbb{N}$. Let

$$q(a) = \arg\min_{q \in Q} |q - a|.$$

From this, we can determine that the quantization error is bounded by

$$\sup_{a \in T} \frac{|q(a) - a|}{a} \leq \frac{1}{K}.$$

Define a prior over the values of $Q$:

$$P(q_{k,b}) = P(k)P(b) = \frac{1}{K} \frac{1}{Z(b+2)(\log_2(b+2))^2}.$$

By direct calculation, we see that $1 = \sum_{k,b} P(k)P(b) = \left( \sum_{b=0}^{\infty} \frac{1}{(b+2)(\log_2(b+2))^2} \right)/Z \leq 1/Z$, therefore $Z \leq 1$.

Now we apply a union bound for Lemma A.3 over values of $t \in Q$. For each $t \in Q$, we set $\epsilon(t) = \delta P(t)$. We have

$$\mathbb{P}\left[ \forall t \in Q : t^2(V + B\Delta) - (B + \Delta C_{\epsilon(t)})t + C_{\epsilon(t)} < 0 \right]$$
$$\leq \sum_{t \in Q} \mathbb{P}\left[ t^2(V + B\Delta) - (B + \Delta C_{\epsilon(t)})t + C_{\epsilon(t)} < 0 \right]$$
$$\leq \sum_{t \in Q} \epsilon(t) = \delta \sum_{t \in Q} P(t) = \delta.$$

Therefore, uniformly for all $t \in Q$, we have

$$t^2(V + B\Delta) - (B + \Delta C_{\epsilon(t)})t + C_{\epsilon(t)} \geq 0 \tag{14}$$

with probability at least $1 - \delta$.

Now we plug in $t = q(t^*)$. Note that $0 \leq b \leq \log_2 \frac{2}{C}$, so we have

$$\log \frac{1}{\epsilon(t)} = \log \frac{1}{\delta P(t)} \leq \log \frac{K}{\delta} + \log(3 + \log_2 1/C) + 2 \log \log_2(3 + \log_2 1/C)$$
$$\leq \log \frac{K}{\delta} + 2 + 2 \log \log 1/C \leq \log \frac{K}{\delta} + 2 + 2 \log \log n \tag{15}$$

Plugging in this quantized value of $t$ to equation 14 and using the quantized error bound, equation 15, we have

$$\frac{(B + \Delta C)^2}{4(V + \Delta B)} \le C + \frac{3}{K}\left(1 + \frac{1}{K}\right)\frac{1}{4}\frac{(B + \Delta C)^2}{(V + \Delta B)}$$
$$\le C(1 + 4/K), \tag{16}$$

where in the second line we chose a $K \ge 6$ so that we have $\left(1 - \frac{3}{K}\left(1 + \frac{1}{K}\right)\right)^{-1} \le 1 + \frac{4}{K}$. Solving equation 16 for $B$, we have the inequality

$$B \le \Delta C(1 + 8/K) + \sqrt{\Delta^2 C^2 (8/K)^2 + 4CV(1 + 4/K)}$$
$$\le \Delta C(1 + 16/K) + 2\sqrt{VC(1 + 16/K)}, \tag{17}$$

where the second line follows from the fact that $\sqrt{x + y} \le \sqrt{x} + \sqrt{y}$. Define $\mathcal{C} = C(1 + 16/K)$. Choosing $K = \lceil 16 \log 1/\delta \rceil$ (which is $> 6$), we have

$$n\mathcal{C} \le \log 1/\delta + 1 + \left[\log(\lceil 16 \log 1/\delta \rceil) + 2 + 2 \log \log n\right]\left(1 + 1/\log(1/\delta)\right) \tag{18}$$

Applying some simplifications to equation 17 and equation 18, we obtain

$$B \le \Delta \mathcal{C} + 2\sqrt{V\mathcal{C}}.$$

with

$$\mathcal{C} \le \frac{1}{n}\left(\log 1/\delta + 4 \log \log n/\delta + 6\right),$$

as desired. $\qquad\square$

## A.2 GENERALIZATION BOUND

**Converting to Prefix Free Codes** Through Kraft-Mcmillan inequality (Kraft, 1949; McMillan, 1956) a binary prefix free code exists if and only if the code lengths $\ell_i$ for the different elements satisfy $\sum_i 2^{-\ell_i} \le 1$. Thus if we have $L(h)$ as the length of a non prefix free code for each $h$, to find the length of a valid prefix code we just need to find $\ell(h)$ such that $\sum_h 2^{-\ell(h)} \le 1$. Using $\ell(h) = L(h)$, the sum would diverge. Instead, consider $\ell(L) = L + 2\log_2(L) + 1$. Computing the sum, $\sum_h 2^{-\ell(h)} = \sum_{L=1}^{\infty} 2^L 2^{-\ell(L)} = \sum_{L=1}^{\infty} \frac{1}{2L^2} = \pi^2/12 < 1$. Thus we can convert any non prefix free code into a prefix free code with using $L + 2\log_2(L) + 1$ bits.

Simultaneously, one can use the prior $P(h) = \frac{12}{\pi^2} 2^{-\ell(h)}$ for any countable hypothesis class, placing higher mass on elements with shorter descriptions, closely related to the Solomonoff prior.

If we know the length of the object ahead of time, then we are free to use a regular code in place of a prefix free code. For a fixed number of parameters $N$ and bits per parameter $b$, we know the length of the code, and thus we can use $\ell(h) \le bN$.

**Weighted Union Bound** Applying Theorem 3.1 to the sequence $R_h(X_k \mid X_{<k})$ with $\delta(h) = \epsilon P(h)$ for each hypothesis individually, the probability that the bound is violated for an arbitrary hypothesis constrained with a union bound $\sum_h \epsilon P(h) = \epsilon$, and therefore Theorem 3.2 holds with probability at least $1 - \epsilon$ (replacing $\delta$ with $\epsilon$ in its expression). The $\log 1/\delta$ in Theorem A.5 becomes $\log 1/\delta + \log 1/P(h) \le \log 1/\delta + \ell(h) \log 2$.

## A.3 PREDICTION SMOOTHING

**Theorem A.6.** *For the categorical negative log likelihood objective $\hat{R}_h = -\frac{1}{n}\sum_{k=1}^{n} \log p_h(X_k \mid X_{<k})$ on $V$ classes and $C \in \mathbb{R}^+$, there exists a prediction smoothed model $p_s(\cdot) = (1 - \alpha)p_h(\cdot) + \alpha/V$ with worst case loss $\sup_{X_k, X_{<k}} -\log p_s(X_k \mid X_{<k}) \le \Delta_s$ that satisfies*

$$\hat{R}_s + C\Delta_s \le \hat{R}_h + C \log V + \sqrt{2C}, \tag{19}$$

*for some value $\alpha(C, V) \in (0, 1)$ (approximately $C/(1 + C)$).*

*Proof.* We have

$$-\log p_s = -\log\big((1-\alpha)p_h + \alpha/V\big)$$
$$\leq -\log p_h - \log\big(1 - \alpha + \alpha/V\big).$$

Noting that $-\log p_s(X_k|X_{<k}) \leq \Delta_s = \log(V/\alpha)$, so adding $C\Delta$ to both sides yields

$$\hat{R}_s + C\Delta_s \leq \hat{R}_h - \log(1 - \alpha + \alpha/V) + C\log(V/\alpha), \tag{20}$$

where the right-hand side is minimized at

$$\alpha = \frac{VC}{(V-1)(1+C)}.$$

Note that $\alpha$ is a *deterministic quantity* that we can compute ahead of time based on the model we are bounding. Therefore we need not pay additional bits for a union bound over values of $\alpha$. Substituting $\alpha$ into equation 20, we have

$$\hat{R}_s - \hat{R}_h + C\Delta_s \leq \log(1+C) + C\log\frac{(V-1)(1+C)}{C}$$
$$\leq (1+C)\log(1+C) + C\log(V/C)$$
$$\leq C\log V + \sqrt{2C},$$

where the last line follows from $(1+x)\log(1+x) - x\log x \leq \sqrt{2x}$ for $x > 0$. The claim follows. $\quad\square$

## B  QUANTIZABILITY FROM THE HESSIAN

We have shown that the quantization gap tends to be quite small in practice, but why is this the case? A more complete explanation of why LLMs generalize would need to explain why they are readily quantizable, not just why they should achieve a small generalization gap if they are quantizable. In this section we attempt to shed light on why there should exist quantized models which achieve low quantization error using a small number of bits per parameter for large models. Here, like in Section 5.2 we focus on demonstrating that these schemes exist, even if they are difficult to find or computationally inefficient to use in practice.

As a starting point in the analysis of many quantization schemes (Nagel et al., 2020), consider the Lagrange remainder form of the quadratic Taylor expansion of the loss around a given solution of the weights $\theta$, with $\hat{\theta}$ being our desired quantization. From this expansion we have

$$L(\hat{\theta}) = L(\theta) + g^\top(\hat{\theta} - \theta) + (\hat{\theta} - \theta)^\top H(\hat{\theta} - \theta)$$

holding with equality for $g$ evaluated at $\theta$ and the Hessian $H$ evaluated at an unknown but fixed point $\xi$ on the linear path between $\theta$ and $\hat{\theta}$, and with no higher order terms. If we use a stochastic rounding algorithm that is unbiased, then the first order term can be neglected as

$$\mathbb{E}[g^\top(\hat{\theta} - \theta)] = 0,$$

and a high dimensional vector $\theta$ ensures the sum will concentrate around the expectation. This leaves the quadratic form with the Hessian. This quadratic form is what many adaptive rounding schemes minimize through the design of their algorithms and in their analysis.

A key property for low precision quantization of the weights (while minimizing the quadratic quantization error) is that the scale of the individual components of the eigenvectors of $H$ do not differ by a large extent. If they do, then the quantization range must simultaneously provide coverage over a large range of values. This criterion is formalized through the notion of incoherence, introduced in Chee et al. (2024), which we briefly present below with a simplification of their more general analysis.

### B.1  INCOHERENCE

A Hessian is $\mu$-incoherent if the eigenvectors in the decomposition $Q\Lambda Q^\top = H \in \mathbb{R}^{N \times N}$ satisfy

$$\forall i, j : |Q_{ij}| \leq \mu/\sqrt{N},$$

and a parameter vector $\theta$ is $\mu$-incoherent if it satisfies $\forall j : |\theta_j| \leq \mu\|\theta\|/\sqrt{N}$. Intuitively this condition can be understood to be stating that the elements are not much more extreme in magnitude than that of an equivalently sized Gaussian random matrix or Gaussian random vector.

A key insight from QuIP (Chee et al., 2024) is that rather than quantizing the weights $\theta$, one should look to quantize the weights after applying a random orthogonal transformation matrix $P \in \mathbb{R}^{N \times N}$. Even as the original weights $\theta$ and eigenvectors $Q$ may be more sharply peaked for certain dimensions, multiplying by a random matrix helps to spread these extreme values across dimensions leading to a more similar range of values and more easy quantization.

Let $w = P^\top \theta$ and likewise $\theta = Pw$. Applying this Gaussian random matrix, the Hessian is transformed: $H_w = P^\top H_\theta P$ and likewise the eigenvectors $Q$ from $H_\theta = Q\Lambda Q^\top$ are also multiplied $Q^\top \mapsto Q^\top P$. If we choose $P$ as a random Gaussian matrix: $\mathcal{N}(0, 1/N)^{N \times N}$, applying a rotation by $Q^\top$ preserves the spherically symmetric distribution. Therefore, the eigenvectors $Q^\top P$ of $H_w$ are $\mathcal{N}(0, 1/N)^{N \times N}$ distributed. Applying a union bound over the Gaussian tail probability of the $N^2$ elements, the maximum absolute value entry of $Q$ is at most $\sqrt{\frac{2\log(2N^2/\delta)}{N}}$ with probability $1 - \delta$ and therefore incoherent with $\mu = \sqrt{2\log(2N^2/\delta)}$. This level of incoherence after applying the random transformation makes for easy quantizability, and as we will see in the next section it has implications on how the quantizability changes with the number of parameters $N$ in the neural network.

## B.2 SCALAR LDLQ

QuIP introduces the LDLQ quantization algorithm which quantizes weights sequentially and autoregressively taking into account how previous quantized values impact the quadratic Taylor expansion of the loss. Applying LDLQ to the entire vector of weights $w$ rather than block by block, one has the following relation on the quantized weights $\hat{w}$. Let $L^\top DL = H_w$ be the LDL decomposition of $H_w = P^\top H_\theta P$, then we can express the quantization of the weights as

$$\hat{w} = \mathcal{Q}(w + (L - I)(w - \hat{w}))$$

where $\mathcal{Q}$ quantizes the weights element-wise with nearest or unbiased stochastic rounding. As $L - I$ is a lower triangular matrix, the full $\hat{w}$ can be quantized sequentially in an autoregressive manner. With this quantization scheme, Tseng et al. (2024) prove that the error of the quadratic in the Taylor expansion satisfies

$$(\hat{w} - w)^\top H(w - \hat{w}) \leq \frac{\mu^2 \sigma^2}{N} \operatorname{Tr}(H^{1/2})^2, \tag{21}$$

where the pointwise quantization error of the scalar quantizer is assumed to be $\mathbb{E}[(\mathcal{Q}(x) - x)^2] \leq \sigma^2$ (see Theorem 4.1 of Chee et al. (2024) applied to block size 1 and a single $N \times 1$ weight matrix), where $\sigma^2$ is a function of the bitrate. For example $x \in [0, 1]$, then a uniform grid would achieve $\sigma^2 = 2^{-2b-2}$ for $b$ bits per parameter. The range of $x$ is not $[0, 1]$ and constraining the number of bits required for more sophisticated schemes requires additional analysis, but for illustrative purposes we will use $\sigma^2 = 2^{-2b-2}$. Putting the pieces together, we have that the difference in loss between the quantized model and unquantized model is

$$L(\hat{w}) - L(w) \leq \frac{2\log(2N^2/\delta)2^{-2b-2}}{N} \operatorname{Tr}(H^{1/2})^2$$

for $H$ evaluated at some point $\xi$ on the linear path between $\hat{w}$ and $w$. Setting the acceptable quantization error $Q = L(\hat{w}) - L(w)$ we can solve for $b$, finding

$$b \leq \log_2\left(\frac{\operatorname{Tr}(H^{1/2})}{\sqrt{NQ}}\right) + (1/2)\log_2\log(2N^2/\delta) - 1/2.$$

Here we see that the number of bits per parameter required depends most heavily on $\log_2\left(\frac{\operatorname{Tr}(H^{1/2})}{\sqrt{NQ}}\right)$.

If the spectrum of $H$ is such that $\operatorname{Tr}(H^{1/2})$ scales slower than $\sqrt{N}$ (such as if $H$ were low rank or has a spectrum the decays sufficiently rapidly), then the number of bits required for quantization at a fixed loss decreases with scale. In the following section we provide some empirical evidence that indeed this is the case, and hence why larger models become more quantizable (even if the quantization scheme is impractical to implement).

### B.3   ESTIMATING $\text{Tr}(H^{1/2})$

To estimate the trace of the square root of the Hessian matrix, $\text{Tr}(H^{1/2})$, we begin by assuming that the Hessian is positive semi-definite (i.e., it contains no negative eigenvalues). The square root of the Hessian, denoted as $S$, can be expressed as:

$$S = \sum_{i=1}^{P} \sqrt{\lambda_i} \phi_i \phi_i^T,$$

where $\lambda_i$ and $\phi_i$ represent the eigenvalues and corresponding eigenvectors of the Hessian, respectively. Consequently, the trace of the square root of the Hessian is:

$$\text{Tr}(H^{1/2}) = \sum_{i=1}^{P} \sqrt{\lambda_i} = n \int_0^{\infty} p(\lambda)\sqrt{\lambda}\, d\lambda,$$

where $p(\lambda)$ is the spectral density function associated with the Hessian's eigenvalues.

A direct computation of the full eigendecomposition to obtain $\text{Tr}(H^{1/2})$ has a computational complexity of $\mathcal{O}(n^3)$, which is infeasible for large models. Instead, we employ stochastic spectral density estimation techniques (Granziol et al., 2018; Papyan, 2019; Ghorbani et al., 2019), which scale linearly with the number of parameters. The key idea involves using the Pearlmutter trick (Pearlmutter, 1994) to efficiently compute Hessian-vector products:

$$\nabla(\nabla L^T v) = Hv,$$

where $v$ is a random vector. This allows us to approximate the trace by leveraging the identity:

$$\text{Tr}(H) = \mathbb{E}[\text{Tr}(vv^T H)] = \mathbb{E}[v^T H v],$$

assuming $v$ has zero mean and unit variance. These stochastic methods are well-established in machine learning (Fitzsimons et al., 2017; Dong et al., 2017).

Building upon the work of Ubaru et al. (2017), we can derive an explicit bound on the estimation of $\text{Tr}(H^{1/2})$ using stochastic Lanczos quadrature (SLQ).

**Theorem B.1.** *Let $\boldsymbol{H} \in \mathbb{R}^{n \times n}$ be a symmetric positive definite matrix with eigenvalues ordered as $\lambda_1 \geq \lambda_2 \geq \cdots \geq \lambda_n$ and condition number $\kappa = \frac{\lambda_1}{\lambda_n}$. For any $\epsilon, \eta \in (0,1)$, if the SLQ parameters satisfy*

$$m \geq \frac{\sqrt{\kappa}}{4} \log \frac{K}{\epsilon} \quad \text{(Lanczos steps)},$$

$$n_v \geq \frac{24}{\epsilon^2} \log \frac{2}{\eta} \quad \text{(Rademacher vectors)},$$

*where $K = (\lambda_{\max} - \lambda_{\min})(\sqrt{\kappa} - 1)^2$, then the output $\Gamma$ of stochastic Lanczos quadrature satisfies:*

$$\Pr\left[\left|\frac{\text{Tr}(\sqrt{\boldsymbol{H}}) - \Gamma}{\text{Tr}(\sqrt{\boldsymbol{H}})}\right| \leq \epsilon\right] \geq 1 - \eta.$$

The proof of this theorem is provided in Section B.6. However, we observe that the bound on the trace provided here is overly conservative for practical purposes. Therefore, we also establish a result demonstrating self-averaging, which shows that the estimator converges to the true value based on a single random vector.

**Theorem B.2.** *For a single random vector $v$, the signal-to-noise ratio of the trace estimator for a matrix $\boldsymbol{H} \in \mathbb{R}^{n \times n}$, where the spectral moments of $\boldsymbol{H}$ do not depend on the matrix dimension, scales as:*

$$\frac{\sqrt{\text{Var}(v^T \boldsymbol{H} v)}}{\mathbb{E}(v^T \boldsymbol{H} v)} = \mathcal{O}(n^{-\frac{1}{2}}).$$

We utilize the CoLA (Potapczynski et al., 2023) library to compute the spectral approximation of the Hessian. This involves leveraging the relationship between the Lanczos $\boldsymbol{T}$ matrix and Gaussian quadrature (Meurant & Strakoš, 2006; Granziol et al., 2019). However, these concepts are highly

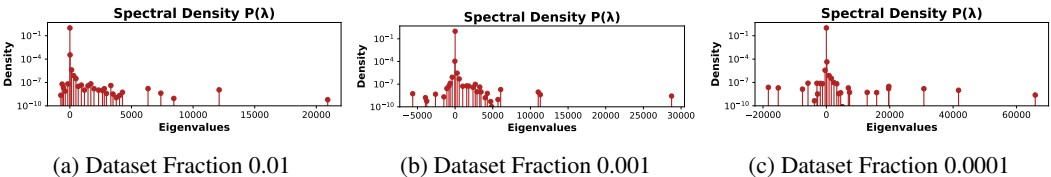

(a) Dataset Fraction 0.01  (b) Dataset Fraction 0.001  (c) Dataset Fraction 0.0001

Figure 4: Spectral density plots of the $70M$ parameter Pythia model trained on varying fractions of the Pile dataset using the same data and random vector seed.

specialized and may not be familiar to all readers. Therefore, we provide a high-level overview without delving into the intricate mathematical details.

Figure 4 illustrates the spectral density of a $70M$ parameter Pythia model trained on different subsets of the Pile dataset (Gao et al., 2020). Specifically, as we decrease the number of training samples—from 1% (Figure 4a) to 0.1% (Figure 4b) and further to 0.01% (Figure 4c)—we observe an increase in the largest eigenvalue and an increase in the mass of negative spectral density. These phenomena are consistent with previous studies on ResNets and VGGs, where spiked Wigner random matrix theory models have been employed to understand such behaviors (Granziol et al., 2022).

Future work aimed at establishing a tighter empirical bound could explore advanced random matrix theory techniques (Bun et al., 2017), potentially utilizing the variance of the Hessian (Granziol et al., 2022). In this study, we adopt a simpler approach by shifting the Hessian spectrum by the magnitude of the largest negative eigenvalue, thereby ensuring a positive semi-definite Hessian and providing a trivial upper bound.

From Figure 4b, we observe that the variance of each estimator remains low and that convergence is achieved with relatively few Lanczos iterations. Additionally, Figure 5 demonstrates that varying the random vector introduces minimal variance, while different data subsets do exhibit some variance, as indicated by the error bars computed over three different seeds (see Figure 5b). For clarity, Figure 5a provides another example of the spectrum with a different seed vector on the same subsampled dataset, showing negligible differences compared to Figure 4a.

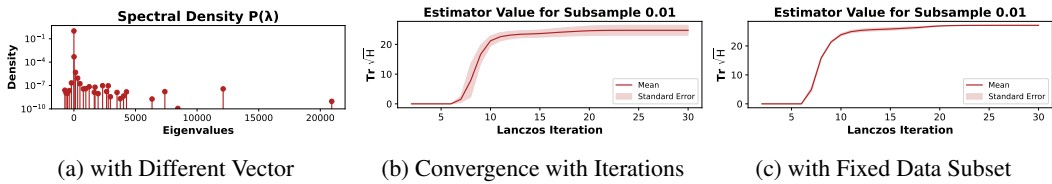

(a) with Different Vector  (b) Convergence with Iterations  (c) with Fixed Data Subset

Figure 5: Comparison of spectral density and $\mathrm{Tr}(\sqrt{\boldsymbol{H}})$ estimations for different subsample sizes and configurations.

With confidence in the accuracy of our estimations for $\mathrm{Tr}(H^{1/2})$ and the Hessian spectrum, we can interpret the implications for model quantization. Despite the high dimensionality and the presence of many distinct eigenvalues, the Hessian spectrum decays rapidly in density. This indicates that $\mathrm{Tr}(H^{1/2})$ grows sublinearly with the model dimension, rather than exhibiting the worst-case linear scaling. Consequently, as model size increases, the ratio $L(h)/D$ is expected to decrease, allowing for a more favorable tradeoff between the bitrate and the quantization gap. This supports the hypothesis that larger models on the compute-optimal frontier are more easily quantizable, thereby contributing to their improved generalization performance.

## B.4 STOCHASTIC TRACE ESTIMATION IMPROVEMENT WITH MODEL SIZE

Here, we provide the proof that for a spectrum independent of model dimension, the stochastic trace estimator has a bigger signal to noise ratio as a function of dimension.

**Lemma B.3.** *Let $\boldsymbol{u} \in \mathbb{R}^{P \times 1}$ random vector, where $\boldsymbol{u}_i$ is zero mean and unit variance and finite 4th moment $\mathbb{E}[\boldsymbol{u}_i^4] = m_4$. Then for $\boldsymbol{H} \in \mathbb{R}^{P \times P}$, we have*

(i) $\mathbb{E}[\boldsymbol{u}^T \boldsymbol{H} \boldsymbol{u}] = \operatorname{Tr} \boldsymbol{H}$,

(ii) $\operatorname{Var}[\boldsymbol{u}^T \boldsymbol{H} \boldsymbol{u}] \le (2 + m_4) \operatorname{Tr}(\boldsymbol{H}^T \boldsymbol{H})$.

*Proof.* For the expectation, we see

$$\mathbb{E}[\boldsymbol{u}^T \boldsymbol{H} \boldsymbol{u}] = \sum_{i,j=1}^P \boldsymbol{H}_{i,j} \mathbb{E}[\boldsymbol{u}_i \boldsymbol{v}_j] = \sum_{i=1}^P \boldsymbol{H}_{i,i} = \operatorname{Tr} \boldsymbol{H}.$$

For the variance, we have

$$\begin{aligned}
\mathbb{E}[||\boldsymbol{u}^T \boldsymbol{H} \boldsymbol{u}||^2] &= \sum_{i,j} \sum_{k,l} \boldsymbol{H}_{i,j} \boldsymbol{H}_{k,l}^T \mathbb{E}[\boldsymbol{u}_i \boldsymbol{u}_j^T \boldsymbol{u}_k \boldsymbol{u}_l^T] \\
&= \sum_{i,j} \sum_{k,l} \boldsymbol{H}_{i,j} \boldsymbol{H}_{k,l}^T [\delta_{i,j}\delta_{k,l} + \delta_{i,l}\delta_{j,k} + \delta_{i,k}\delta_{j,l} + m_4\delta_{i,j,k,l}] \\
&= (\operatorname{Tr} \boldsymbol{H})^2 + (2 + m_4) \operatorname{Tr}(\boldsymbol{H}^2),
\end{aligned}$$

whence (ii) follows.

Let us consider the signal to noise ratio for some positive definite $\boldsymbol{H} \succ c\boldsymbol{I}$

$$\frac{\sqrt{\operatorname{Var}[\boldsymbol{u}^T \boldsymbol{H} \boldsymbol{u}]}}{\mathbb{E}[\boldsymbol{u}^T \boldsymbol{H} \boldsymbol{u}]} = \sqrt{2 + m_4} \sqrt{\frac{\operatorname{Tr} \boldsymbol{H}^2}{\operatorname{Tr}^2 \boldsymbol{H}}} = \sqrt{\frac{2 + m_4}{P}} \sqrt{\frac{\langle \lambda^2 \rangle}{\langle \lambda \rangle}} \tag{22}$$

where we denote the mean eigenvalue $\langle \lambda \rangle$ and the mean square eigenvalue similarly. □

*Remark* B.4. Note that $m_4$ is 3 for the Gaussian case and 1 for the Hutchinson trace estimator where the entries are $\pm 1$ with probability half, which justifies its use.

## B.5 Deriving the impact of Low precision Lanczos

Consider a number taken from our Hessian matrix $a_{i,j}$, which can be represented as $(-1)^s 2^e s$. As the exponent for FP16 has 5 bits, it has a range of $2^5 - 1$. Since the exponent is always integer, there is no loss of information in the range. This means the error is in the significand, which has 6 bits after the 1. Thus, we have $\epsilon = 10^{-7}$.

Then, we see that

$$\begin{aligned}
\tilde{\boldsymbol{H}} &= \begin{bmatrix}
a_{1,1}(1 + \mathcal{N}(0,\epsilon)) & a_{1,2}(1 + \mathcal{N}(0,\epsilon)) & \cdots & a_{1,n}(1 + \mathcal{N}(0,\epsilon)) \\
a_{2,1}(1 + \mathcal{N}(0,\epsilon)) & a_{2,2}(1 + \mathcal{N}(0,\epsilon)) & \cdots & a_{2,n}(1 + \mathcal{N}(0,\epsilon)) \\
\vdots & \vdots & \ddots & \vdots \\
a_{m,1}(1 + \mathcal{N}(0,\epsilon)) & a_{m,2}(1 + \mathcal{N}(0,\epsilon)) & \cdots & a_{m,n}(1 + \mathcal{N}(0,\epsilon))
\end{bmatrix} \\
&= \boldsymbol{H} + \begin{bmatrix}
a_{1,1}\mathcal{N}(0,\epsilon) & a_{1,2}\mathcal{N}(0,\epsilon) & \cdots & a_{1,n}\mathcal{N}(0,\epsilon) \\
a_{2,1}\mathcal{N}(0,\epsilon) & a_{2,2}\mathcal{N}(0,\epsilon) & \cdots & a_{2,n}\mathcal{N}(0,\epsilon) \\
\vdots & \vdots & \ddots & \vdots \\
a_{m,1}\mathcal{N}(0,\epsilon) & a_{m,2}\mathcal{N}(0,\epsilon) & \cdots & a_{m,n}\mathcal{N}(0,\epsilon)
\end{bmatrix}.
\end{aligned}$$

Now under certain assumptions on the elements of the perturbation matrix (essentially the $a_{i,j}$ does not vary too wildly or have wild dependencies) this becomes a Gaussian orthogonal ensemble (GOE) again. Then using the Frobdyenius Norm, we see that the spectral width will be of order $\epsilon\sqrt{\langle \lambda^2 \rangle}$, which depends on the square root of the average eigenvalue squared of $\boldsymbol{H}$. Anything within this will be noise. This is because $\sum_{i,j} a_{i,j}^2 = P\langle \lambda^2 \rangle$. An obvious upper bound of this would be $\epsilon\lambda_1$ but this will likely be super loose. Note that the vast majority of the already broadened spectrum is already very close to zero, so we would expect this to be even more extreme for the unbroadened version. A better strategy might be to sample the noisy version of $a_{i,j}^2$ perhaps using the diagonal approximation, and note that in expectations we expect the square to be $(1 + \epsilon^2)$ the size of its non noisy counter part, which gives an an estimation equation

$$\sqrt{\frac{P\epsilon^2 \sum_k^N a_{i,j}^2}{N(1 + \epsilon^2)}}.$$

### B.6 Stochastic Lanczos Quadrature Proof

**Theorem B.5.** *Consider a symmetric positive definite matrix $\boldsymbol{A} \in \mathbf{R}^{n \times n}$ with eigenvalues enumerated in reverse order of size $\lambda_1 \geq \lambda_2 \cdots \geq \lambda_n$ and condition number $\kappa = \frac{\lambda_1}{\lambda_n}$. For $\epsilon, \eta \in (0, 1)$ and SLQ parameters satisfying*

*(i)* $m \geq \dfrac{\log \frac{K}{\epsilon}}{2 \log \frac{\sqrt{\kappa}+1}{\sqrt{\kappa}-1}} >= \dfrac{\sqrt{\kappa}}{4} \log \frac{K}{\epsilon}$ *Lanczos steps*

*(ii)* $n_v \geq \frac{24}{\epsilon^2} \log \frac{2}{\eta}$ *Rademacher vectors,*

*where $K = (\lambda_{max} - \lambda_{min})(\sqrt{\kappa} - 1)^2$. The output $\Gamma$ of stochastic lanczos quadrature is such that*

$$\Pr\left[ \left| \frac{\text{Tr}(\sqrt{\boldsymbol{A}}) - \Gamma}{\text{Tr}(\sqrt{\boldsymbol{A}})} \right| \leq \epsilon \right] \geq 1 - \eta \tag{23}$$

*Proof.* The proof follows trivially from Ubaru et al. (2017), where we simply take the more general proof and instead of the general function $f(\boldsymbol{A})$, we take $f(x) = \sqrt{x}$. The second inequality for $m$ is directly from the paper, but the tighter bound is also available just buried. $\qquad\square$

The proof sketch goes as follows. We bound the error from the Gauss quadrature rule. We start with a function analytic in the interval $[-1, 1]$. Knowing that the Gauss quadrature rule is exact for any polynomial up to degree $2m + 1$, we bound the sum from $2m + 1$ to infinity using Cauchy-Schwarz. We use results from Chebyshev coefficients, symmetry and the interval boundaries to get

$$|I - I_m| \leq \frac{4\sqrt{\lambda_1}}{(\rho^2 - 1)\rho^{2m}},$$

where $\rho$ is the sum of the major and minor axis of the Bernstein ellipse. We shift the spectrum so that it is in the interval $[-1, 1]$, e.g this implies the factor of $\frac{\lambda_1 - \lambda_n}{2}$. The shifted function is not analytic for $\alpha = -\frac{-\kappa+1}{\kappa-1}$, so this will serve as our major axis. Now as $\frac{x^2}{a^2} + \frac{y^2}{b^2} = 1$ and the focus is $1 = \sqrt{a^2 - b^2}$, where we take our major axis $a$ in this case to be $\alpha$. We then have our rate of convergence $\rho = a + b$ through some algebra to be $\frac{\sqrt{\kappa}+1}{\sqrt{\kappa}-1}$. This gives us the value of $K$. This is combined with the error of the trace estimator from Roosta-Khorasani & Ascher (2015) and Cauchy-Schwartz to obtain the final result.

