# OpenReview forum: "Compute-Optimal LLMs Provably Generalize Better with Scale"
_ICLR.cc/2025/Conference — ICLR 2025 Poster_

### Official Review · Reviewer_dkJs · 2024-10-31

**Soundness:** 3
**Presentation:** 3
**Contribution:** 3
**Rating:** 6
**Confidence:** 2

**Summary:**

Building on an empirical version of Freedman's inequality - a concentration bound for martingales - the authors derive generic generalization bounds for large language models. These bounds are obtained in a PAC setting and do not take into account the training of the neural network considered or the model's precise architecture. They are empirical, meaning that they can be computed in practice with a limited computational budget, and are shown to hold for a single architecture and dataset. The authors then argue that these bounds get tighter as the number of parameters increases by analyzing information accumulation within the model.

**Strengths:**

**On the overall presentation of the paper.**

Overall, the paper is well written. The contributions are explained in great detail and are not overstated. Most of the mathematical and theoretical background needed to understand the article is provided in the main text. Figures are clear and well designed.

**On the empirical version of Freedman's inequality.**

This theorem and its consequences for empirical loss minimization are the central contributions of this article. I believe this concentration bound to be valuable and interesting, mostly because of its empirical variance part which can be approximated as the solution of an optimization problem on a discrete real-valued grid.

**On the derived generalisation bounds.**

The authors combine this bound with the smoothing technique of the model of Lotfi et al. (2024a), which in a nutshell convolutes the model with a uniform distribution to smoothen the prediction function, hence incurring an error but providing an interesting proxy of the empirical unsmoothed loss. Theorem 3.4 then combines this technique with the empirical Freedman's inequality to obtain a quite generic (i.e. model and training independant) generalization bound.

**Weaknesses:**

**On the comparison with related work.**

While the authors cite some related work on generalization bounds for deep models, their literature review lacks in my opinion a precise comparison with previous approaches. In particular, I would enjoy both a theoretical, mathematical and empirical comparison of the derived bounds with other bounds present in the literature - see their section 6.

**On the tightness of the generalization bounds.**

From what I understand, the generalisation bounds display a strong linear dependency in the number of parameters - at least in the form presented in theorem 3.4. The authors later on argue that this bound can be improved, by improving on the inequality $L(h) \leq bN \log 2$. While interpretable and empirical, these bounds display an extreme dependence in the number of parameters, which is undesirable in the case of LLMs.

**On empirical evaluation.**

The validity of the author's claims is assessed on a single LLM on a given dataset. I am unsure whether this evaluation is sufficient. I have little experience in the evaluation and training of LLMs, hence my analysis of this part is limited. I will revise my judgement based on other reviews and discussions with the authors.

**On Section 5.**

My principal concern lies in the arguments made in section 5. The authors argue that while their bound is linear in the number of parameters in the form presented in Theorem 3.4, one can refine this bound and obtain a bound which is independent of $N$ the number of parameters. This section is not clear and hard to understand --- I stress that I am not familiar with prequential coding and might hence not be the targeted audience here. In the end, I do not understand how the authors conclude from Equation $(6)$ that their generalization bound improves with the number of parameters. From what I understand, the authors claim that their bound improves as the number of parameters increases only if the size of the dataset increases as well. If this is the case, I believe that the significance of their results is overclaimed.

**Supplementary comment.**

I also raise to the other reviewers and chairs attention that this article relies on the smoothing technique designed by Lotfi et al. (2024a). The validity of the results of this paper has been questioned during review (see https://openreview.net/forum?id=GY1fKFXG5i&noteId=s9ciWh5QAI) leading to a rejection of this paper at another conference. While I am unable to assess whether this interrogations are justified, and if the possible errors in this paper might carry over to the results exposed here, I believe that this information should be shared among reviewers and chairs.

**Questions:**

* Could you please provide supplementary thorough explanations of section 5 ? I am puzzled by the arguments made here.

* In regard of this last question, could you consider including a supplementary part which gives more background on these ideas ?

* Could you give mathematical details on how your proof techniques rely on the results of Lotfi et al. (2024a) ? In the worst case, if these results were to be false, would your analysis still hold ?

---

> ### Author Response · Authors · 2024-11-21
>
> We thank you for your detailed review. We address your questions one by one.
>
> For the most closely related work see Lotfi et. al (b, 2024) which we cite and compare our generalization bounds to. As stated at the end of the background section and additionally in the related work section, this work uses Azuma’s inequality producing bounds that scale $\sqrt{C} \log V$ whereas our empirical Freedman inequality based bounds scale as $C\log V + \sqrt{C}\Sigma$. $C\log V$ dominates the other contributions to the bounds and as $C\approx 1/9$, our bounds are roughly 3x tighter than those derived using Azuma’s inequality for the same sized model and dataset.
>
> You mention an ‘extreme’ dependence on the number of parameters of the model. One of the crucial insights that we present here not previously explored in other work is that despite the conventional wisdom, this linear dependence (in the case of using quantization) is not that extreme due to the way that large models are scaled up in both model size and data size. The ratio of parameters to data points is a constant along the compute optimal frontier, and the decreasing information content per parameter means that this ratio becomes even smaller with scale. We do not see how this forms substantive critique of the paper when competing alternatives do not exist within the research literature. If other approaches had been demonstrated on similarly sized LLMs with a superior scaling that would be a different story, but this is not the case.
>
> As mentioned to another reviewer, the evaluation on just the pythia family of models is by necessity. In order to perform the analysis and evaluation of generalization bounds that we do in the paper, we need to have access to both the model checkpoints and unbiased random samples from the data that the model was trained on. To the best of our knowledge Pythia is the only set of models spanning multiple orders of magnitude in parameter count where this is the case. Almost all open source LLMs do not provide their training data, and without that training data we cannot conduct the analysis we do here. While it would be great if additional model families were available that is not currently the case. This is a theory paper and we do not consider it within the scope of such a project to undertake the training of multi billion parameter LLMs. Respectfully, we do not consider this to be a fair reason to advocate for rejecting the paper.
>
> In section 5 we make the argument based on the functional form of the scaling law and prequential coding that the information content in the model grows sublinearly in the number of parameters (or the number of data points) in the compute optimal regime. For an overview of section 5, the argument can be summarized as follows. Using prequential coding, the model and data can be coded jointly in a number of nats equal to the cumulative sum of the NLL values over the course of training (for a modern reference, see e.g. Blier and Ollivier 2018). Given the final model, the data alone has a code length equal to the sum of loss values of the final model, and by assumption this is well described by the scaling law. Thus we can use this value to estimate the complexity of the data given the final model. Through the symmetry of information property of Kolmogorov complexity, the difference between the two (up to logarithmic factors) is equal to $K(M)$, the complexity of the model.
>
> Getting from equation 5 to equation 6, we simply evaluate the functional form of the scaling law $R(N,D) = E+AN^{-\alpha}+BD^{-\beta}$. $\sum_{k=1}^D R(N,k) = D(E+AN^{-\alpha}) + \sum_{k=1}^d B k^{-\beta}$. Subtracting the two terms, $\sum_{k=1}^D R(N,k) - DR(N,D) = B(\sum_{k=1}^D k^{-\beta}-D^{1-\beta})$. Asymptotically, $\sum_{k=1}^D k^{-\beta} \sim \frac{1}{1-\beta}D^{-\beta}$ therefore the difference is $O(D^{1-\beta})$. Does this clear up your understanding of the section? And if not, please let us know which parts you don’t follow so we can clarify them, particularly as you listed this as a principal concern in your review.
>
> “From what I understand, the authors claim that their bound improves as the number of parameters increases only if the size of the dataset increases as well. “
> Yes. The information content in the model grows sublinearly while the dataset size grows linearly, leading to a shrinking complexity term and thus improving bounds. We believe this is a highly significant result as such observations have not been made in the literature.

---

> > ### Author Response · Authors · 2024-11-21
> >
> > Regarding your supplementary comment to the other reviewers and chairs, while we appreciate the concern, we believe there may be a misunderstanding of the points there.
> > Firstly, the question from a reviewer in the openreview discussion for Lotfi et al. (2024a) that you linked was not about the validity of prediction smoothing, it was about whether the sequence chunk sampling employed constitutes IID sampling from the data. In our work we do not require an IID assumption and instead employ token level martingale based analysis. Secondly, the authors showed the iid assumption was valid, and that paper was subsequently accepted in ICML 2024, with discussion of the IID assumption in publicly available openreview for that submission https://openreview.net/forum?id=GY1fKFXG5i&noteId=s9ciWh5QAI. Thirdly we do not rely on any theorems or results for prediction smoothing from Lotfi et al. (2024a), and the relevant results we prove directly in the appendix (theorem A.6).
> >
> > We hope you may consider raising your score in light of our response, we are happy to answer any additional questions you may have.

---

> ### Comment · Reviewer_dkJs · 2024-11-21
>
> Thank you for addressing my questions in great detail.
>
> - Concerning the dependence in parameters, I respectfully stand by my opinion: I do believe that the dependence is extreme, and hence question the quality of this bound. Its main strength in my opinion lies in the fact that it is empirical.
> - I do agree with your point on the evaluation, given that your explanation is satisfactory and that this has not raised major concerns in other reviews.
> - The discussion concerning section 5 is of little help and seems to require theoretical elements with which I am not familiar. Other reviewers have also stressed a lack of clarity: this should raise your attention on this point. Please improve the clarity of this section.
> - Concerning the scaling of the bound, I stand by my point: I still believe that the title of the paper is clearly claiming something stronger than what you show. The fact that under certain assumptions, the complexity-to-datapoints ratio tends to $0$ is a well studied fact in learning theory and is not surprising - even though your proof techniques are novel. If you consider a linear model with sparsity growing at a sufficiently slow speed, such a result would hold, but I doubt that this would be consider as a proof of "larger models generalising better".  From the title, I would have expected that for a fixed datasize, increasing the number of parameters leads to better performance - see for instance http://proceedings.mlr.press/v97/brutzkus19b/brutzkus19b.pdf or https://arxiv.org/abs/1810.02032 for the analysis of such a case.
> - Concerning my concerns with related work of Lotfi et al. (2024a), I thank you for the clarity of your response. I had missed the second submission of this article and have reviewed the discussion. This fully answers my question.
>
>  To sum up, my main concerns lie in (1) the strength of your generalisation bounds, (2) the clarity of your work and (3) the choice of title. To make my comment actionable, I suggest the following:
> - To solve problems (1) and (3), please reconsider the strength of your results by changing the title and maybe amending your article to include a discussion of what I consider to be weaknesses.
> - To solve problem (2), I would suggest a rewriting of section 5 which would include more intuitions and explanations, and the addition of a supplementary that gives some background.
>
> I am not able to raise my score yet but am in no way hostile to raising it during the rest of the discussion period. I will remain active and discuss these issues, since I believe this work has to be improved but builds on solid ground.

---

> > ### Author Response · Authors · 2024-12-02
> >
> > We thank you for your thoughts and willingness to take into account our response.
> >
> > We are on board with the fact that one can find classical learning theory results on particular constructions where generalization improves as model complexity increases in a particular way while the dataset also increases. However we feel that the generalization gap decreasing when the number of parameters in an LLM and the number of tokens that they are trained on are scaled equally is quite interesting and significant and still somewhat surprising in the context of LLMs.
> >
> > On the other hand we do not intend to claim that our bounded generalization gap decreases when only the number of LLM parameters is scaled up, and if our title gives this impression this is not our intention. As mentioned for the other reviewer who had a similar concern about the title overclaiming on the results of the paper we would be happy to commit to changing the title to something along the lines of "Compute Optimal LLMs Provably Generalize Better with Scale” which less ambiguously is in line with the results of our paper.
> >
> > With regards to the empirical evaluation and comparison to other methods in the literature, below we have produced a table comparing the results of our bounds vs equivalent bounds derived from Azuma’s inequality, employed in Lotfi et al (b, 2024). Leveraging our empirical Freedman concentration inequality, we achieve a substantially lower bounded loss value over equivalent Azuma based bounds across model sizes. In addition to the theoretical discussion, we will add this table to the appendix.
> >
> > | Model Size             | 70m   | 160m  | 410m  | 1b    | 1.4b  | 2.8b  | 6.9b  |
> > |---------------------|-------|-------|-------|-------|-------|-------|-------|
> > | Train Loss         | 3.79  | 3.14  | 2.74  | 2.40  | 2.29  | 2.15  | 1.95  |
> > | Our Bound       | 5.52  | 4.99  | 4.88  | 4.54  | 4.42  | 4.23  | 4.06  |
> > | Azuma Bound  | 6.71  | 6.21  | 6.10  | 5.78  | 5.66  | 5.49  | 5.33  |
> >
> > Regarding clarity of the exposition, we have reworded the statement of theorem 3.1 to make it less ambiguous (see the response to wrVe). For section 5, we commit to improving the clarity following our discussion here, adding additional intuition for the argument and approach that we take. While the argument invokes tools from less commonly seen areas (such as algorithmic information theory), we hope that the approach can be insightful to the broader community even if it requires accommodating some additional background. We hope you can consider our commitment to these changes in your final assessment. It could be a pity to go through a whole new round of review for changes that, while important, can be straightforwardly addressed. We also hope we have now highlighted the significance and timeliness of these results throughout the course of the discussion.

---

> > > ### Comment · Reviewer_dkJs · 2024-12-02
> > >
> > > Thank you for your answers. I will raise my score. "Compute Optimal LLMs Provably Generalize Better with Scale" seems to be a good title.

---

### Official Review · Reviewer_wrVe · 2024-11-02

**Soundness:** 3
**Presentation:** 1
**Contribution:** 3
**Rating:** 6
**Confidence:** 3

**Summary:**

The authors present a new upper bound on the single-token generalisation error of language models. Evaluations of this bound are compared with selected empirical data from the training history of a language model.


EDIT: I have raised the score, on the understanding that the authors will amend the title as discussed and significantly clarify the exposition of the theorems, including to clearly state that the 'with probability 1- \delta' refers to the sampling of both the X_k and Y_k.

**Strengths:**

The authors' main result is interesting: an apparently novel generalisation bound which can be evaluated from data. The comparisons they make between their bound and empirical data is also of interest. Hopefully this bound will be useful in future for more efficiently guiding the development of AI models

**Weaknesses:**

1. I am left wondering whether this is really an appropriate venue for this work. Proving the results stated in the actual paper requires 10 pages of dense technical appendices introducing a wealth of additional lemmas, theorems and corollaries which are not reviewable in the timeframe. The results themselves do not appear to be standard, and I feel would benefit from a proper presentation in a format, such as a journal, which gives sufficient space for a detailed investigation of the technical details.
2. The statement of several of the main results do not appear to be complete, and require extra clarification. In particular, the authors should take care to ensure that all notation is defined.
    - Does the 'with probability $1 - \delta$' in Theorems 3.1, 3.2, and 3.4 refer to the sampling of $X_k$, $Y_k$, or both? If both, the authors should clarify how their results on the *joint* probability of $X_k$ and the fictitious random variable $Y_k$ can be connected to standard generalisation bounds which look at the probability with respect to $X_k$. The random variables over which probabilities are being measured should be explicitly stated in all results.
    - The random variables $Y_k$ implicitly appear in the result of Theorem 3.4 (through $\Sigma$), but are not defined in the statement of the theorem.
    - In Theorem 3.1, what is $\mathcal{F}_k$? I expect this is some form of filtration, but it is undefined.
    - In Theorem 3.1, ${X_{k}}$ are ${\mathcal{F}{k}}$-measurable, while $Y_k$ are $\mathcal{F}_{k-1}$-measurable. Given the ambiguities of the notation, it is unclear what this distinction actually refers to.
    - The term 'NLL' is frequently used throughout, but is never defined.
3. In Theorems 3.1, 3.2 and 3.4, the size of the finite set $K$ is a key component of the 'complexity' measure $\mathcal{C}$. Yet, $K$ itself seems to be just a technical construction in the proof, rather than something fundamental to the model. The authors should clarify what role this set plays, and why it should be considered as part of the complexity of the model, rather than simply an artefact of the proof. They should also indicate how this set can be chosen in practice (since the bounds are supposed to be computable). I see from Section 4 that they use 1000 points, but offer no indication of why this choice was made.
4. Additional random variables $Y_k$ are introduced and used in most of the main results. The authors state after Theorem 3.1 that they take this to be the mean of the model NLL, but I do not see what this can be guaranteed to satisfy the requirement that $Y_k - X_k > -\Delta$ (and how is the value of $\Delta$ chosen for this setup?). In fact, why not just take $Y_k = X_k - \Delta + \epsilon$ for some fixed value of $\Delta$ and a small value of $\epsilon > 0$? I do not see anything in the statement of the theorem preventing this, and it appears to optimise the bound.
4. The authors only compare their computable bound to empirical data for selected so-called 'compute-optimal' checkpoints. Why is this choice made? How does the bound compare for other checkpoints? This does not appear to be a requirement of the theorems, and so the rationale should be made explicitly clear.
5. I am not convinced that the results presented by the authors justify the claim in the title. It seems to me that, in fact, their results show that increasing the size of the model will in general *increase* their generalisation bound, by increasing the complexity term. In fact, the authors themselves appear to advocate for compressing and quantising models. The authors should explicitly justify such a strong claim in the paper, or change the title to better reflect the results they present.

**Questions:**

Please see the questions in the Weaknesses box.

---

> ### Author Response · Authors · 2024-11-21
>
> We thank you for your detailed and thoughtful review. We’d like to address your questions in turn and hopefully clear up any questions you may have.
>
> Regarding the main mathematical results of the paper we would like to clarify a few things that may not be immediately apparent. The random variables $Y_k$ serve as a proxy for the conditional mean $\mathbb{E}[X_k|\mathcal F_{k-1}]$. We don’t know the mean (because we want to be able to empirically evaluate the bounds) but in its place we can substitute any deterministic function of quantities that are already known (at time k-1), or in other words $Y_k$ is $\mathcal F_{k-1}$ measurable. As $Y_k$ can be any $\mathcal F_{k-1}$ measurable function, this is why it has no additional definition in theorem 3.1.
>
> As an example, one choice of $Y_k$ and the one we use in evaluating our bounds is choosing $Y_k$ as the mean NLL for token k according to the model taken over samples from the model distribution (conditioned on tokens <k), whereas $X_k$ is the the model NLL for the token drawn from the actual data distribution. Crucially, this model mean can be computed without knowing what token will come next. This choice of $Y_k$ is implicit in theorem 3.2 and 3.4, though we recognize that split between the main text and the appendix this detail may be less forthcoming, and perhaps especially so since in contrast to theorem 3.1, in 3.2 and 3.4 $X_k,Y_k$ are the token k and the model sampled token k. To address this we will add a small amount of additional exposition for this point to appendix A.5 and to theorem 3.2 in the main text.
>
> Thus the with probability $1-\delta$ statement refers to the sampling of the token sequence ($X_k$ in theorem 3.4), from which both the per token loss values and the model mean loss values can be derived.
>
> On the finite set $K$, your observation is appropriate. The discrete set $K$ is chosen to enable optimization over the scale variable $t$ in Theorem A.5. In the proof of A.5 (“Inequality Sketch”) we sketch out how the proof would look if one could freely optimize this variable without regards to making an additional union bound, and why the union bound is necessary. The optimal scale for $t$ depends on how closely $X_k$ concentrates to the mean, however theorem A.3 holds for a single value of $t$ rather than $t$ itself being a random variable. To circumvent this one can quantize the values of $t$ and optimize over them. At $|K|=1000$ the contribution to the complexity term the $(1/D)\log |K|$ factor (see line 264) is completely inconsequential in comparison to the term depending on the model parameters. $|K|=1000$ was chosen not to be larger so as not to slow down the evaluation of the bounds by too much and at that size $\Sigma$ has already converged to sufficient precision. We will add these details to the appendix on your suggestion.
>
> On $Y_k$ and $\Delta$. As the NLL is strictly positive on the finite token distribution, $Y_k-X_k > -X_k >-\Delta$ where $\Delta$ is an upper bound on the per token NLL. The per token NLL is bounded in theorem 3.3 by introducing label smoothing, and $\Delta$ is optimized to be $\log V+\log((1+C)/C)$, slightly larger than random guess NLL of $\log V$. The reason we cannot take $Y_k=X_k-\Delta+\epsilon$ is that then $Y_k$ would not be a $\mathcal F_{k-1}$ measurable random variable and instead would be a $\mathcal F_k$ measurable random variable (it would be a function of unknown information and thus random even given context <k).
>
> We evaluate our generalization bounds on the compute optimal frontier (shown in figure 1). Though large models trained by companies can and do sometimes differ in this choice due to other reasons, the largest and most expensive runs tend to be close to the compute optimal curve (see e.g. LLAMA 3 405B). Abstractly one can consider what would happen if models were trained on far more data or far less data than they do currently, but practically speaking model size and dataset size are scaled together. It is our view that the choice of how compute is expended between these two is a critical component of the algorithm, and thus evaluating the algorithm that produces LLMs requires respecting this choice. In this regard we break with prior work on generalization bounds and attending to this aspect provides motivation for the current work. For checkpoints trained with more data than optimal, their bounds will be tighter in a predictable way and likewise looser for checkpoints trained with less data.
>
> -- continued --

---

> > ### Author Response · Authors · 2024-11-21
> >
> > Downstream of this choice to pay attention to how things change as the models are scaled up in compute is the claims we make that larger models generalize better. In this case large can be understood as referring to compute. Even if referring to parameters what is meant by a large model is one that is trained on an amount of data appropriate to its size.
> >
> > We strongly believe that ICLR is an appropriate venue for this work. So far the only theoretical work on finite hypothesis generalization bounds for large language models has appeared at similar machine learning conferences, Lotfi et. al (a,b, 2024), cited in our submission. Given the significance and timeliness of this work, our clarifications, and the additional context, we hope you can consider revising your assessment.

---

> > ### Comment · Reviewer_wrVe · 2024-11-26
> >
> > Thank you for responding to my comments.
> >
> > If I understand your comments correctly, the probability in the bound in Theorem 3.1 is taken only over the X_k random variables, implying that it is itself still a random variable since it depends on Y_k. I strongly suggest that the authors take another look at how this result is presented to clarify this point.
> >
> > Thank you also for clarifying that the presence of |K| in the 'model complexity' term is simply an artefact of the analysis. It is interesting that the term arising in each bound from the second term of ∑ is not scaled by the 'model complexity' term, although this does not appear to be something the authors comment on.
> >
> > Ultimately, I believe that once the presentation is fully and carefully clarified the theoretical results themselves will be interesting enough for me to increase the score (although I would have expected more thorough empirical results to significantly increase it). However, I cannot justify doing so at present because I do not believe the title is an accurate representation of the work. The theoretical results do not appear to agree with the strong and narrow claim made in the title (the title suggests just increasing the size of the model will improve performance while the results paint a more nuanced picture; the authors' interpretation of 'larger' seems to contain too many 'terms and conditions' to be justifiable), and the empirical results are certainly much too limited to justify the title.

---

> ### Author Response · Authors · 2024-12-02
>
> We thank you for your engagement and thoughtful suggestions.
>
> Returning to theorem 3.1, the randomness is over $(X_k,Y_{k})_{k=1}^n$ and so the inequality is not random. We believe that we have not stated in any of the relevant theorems that the randomness is only over $X_k$ though we recognize the ambiguity.
>
> To clarify the statement of the theorem, we have reworded it as: (forgive the spacing as OpenReview's latex rendering is somewhat idiosyncratic)
>
> "Let $(X_k)_{k=1}^n$
>
> and $(Y_k)_{k=1}^n$
>
> be sequences of random variables adapted to a filtration
> $(F_k)_{k = 0}^{n-1}$,
>
>  where each $X_k$ is $F_k$-measurable and each $Y_k$ is $F_{k-1}$-measurable. Assume the scaled differences $A_k = (Y_k - X_k)/\Delta$ are bounded below by $-1$ for some $\Delta > 0$. Let $K$ be a finite subset of $(0,1)$. Then, with probability at least $1-\delta$ over the probability space filtered by $(F_k)_{k = 0}^{n-1}$ …"
>
>  We believe this clears up any impreciseness in the statement of the theorem and we hope this is to your satisfaction.
>
> In terms of the title, while we have a different perspective we can understand how it could be interpreted as stating something different from what we had intended. If you're willing to raise your score to support acceptance, we would be happy to commit to changing the title to something along the lines of "Compute Optimal LLMs Provably Generalize Better with Scale
> ” which less ambiguously is in line with the results of our paper.
>
> For further empirical validation, we have added a table comparing the results of our bounds vs using bounds derived from Azuma’s inequality, employed in Lotfi et al (b, 2024). Leveraging our empirical Freedman concentration inequality, we achieve a substantially lower bounded loss value over equivalent Azuma based bounds, nearly halving the gap from the training loss across the different model sizes.
>
> | Model Size             | 70m   | 160m  | 410m  | 1b    | 1.4b  | 2.8b  | 6.9b  |
> |---------------------|-------|-------|-------|-------|-------|-------|-------|
> | Train Loss         | 3.79  | 3.14  | 2.74  | 2.40  | 2.29  | 2.15  | 1.95  |
> | Our Bound       | 5.52  | 4.99  | 4.88  | 4.54  | 4.42  | 4.23  | 4.06  |
> | Azuma Bound  | 6.71  | 6.21  | 6.10  | 5.78  | 5.66  | 5.49  | 5.33  |
>
> We really appreciate your support and suggestions.  It would be a pity to have to go through a whole new review cycle for changes that, while important, can be straightforwardly resolved.

---

### Official Review · Reviewer_y45d · 2024-11-03

**Soundness:** 3
**Presentation:** 3
**Contribution:** 3
**Rating:** 8
**Confidence:** 3

**Summary:**

This work provides a novel generalization bound for LLMs consisting of three components depending on the ratio of the number of parameters to tokens, loss variance, the performance gap from model quantization. The paper first provides a new martingale concentration inequality that can be evaluated empirically. The rest of the components of the bound come from loss variation, a smoothing constant, and quantization gap, all of which can be computed empirically. Experiments show that the obtained bounds are very close to evaluated loss functions in LLMs. The paper also provides an alternate bound based on information accumulation in LLMs using prequential coding, which is a less empirical bound. Nevertheless, gives insight on model complexity with changing model scale while keeping the ratio of dataset to parameters constant. Overall, the paper provides novel generalization bounds for LLMs with interesting insights to the working of LLMs and how they vary with scale.

**Strengths:**

- The paper presents two interesting generalization bounds for LLMs, one of them can be empirically calculated and shows results close to actual loss obtained in experiments. The other bounds gives insight on generalization bounds in LLMs with increase in model size with constant data to parameters ratio.
- For computing the first bound, the model derives a new concentration inequality based on the Freedman’s inequality that can be empirically computed and is likely to be of interest in future works.
- The empirical evidence provided in the paper showing that it is close to the obtained bounds in interesting.

**Weaknesses:**

- It would be good to improve the readability of the paper a bit by distilling down the mathematics and giving an intuition for both the bounds in summary.
- It would be better to provide the empirical evidence after both the bounds instead of providing it in between. E.g., some plots are provided in Sec. 5, whereas the empirical evidence is given in Sec. 4.

**Questions:**

- A general question: Are the empirical bounds extendable to LLM performance with inference-compute scaling?, e.g. in [1]

[1] Sardana, Nikhil, et al. "Beyond chinchilla-optimal: Accounting for inference in language model scaling laws." arXiv preprint arXiv:2401.00448 (2023).

---

> ### Author Response · Authors · 2024-11-21
>
> We really appreciate your supportive and thoughtful review!
> We have taken your feedback and will incorporate additionally summary for the bounds we derived in the revised version of the main text. For the placement of figure 2 and section 4 between section 3 (deriving the bounds) and section 5 (compressibility and information growth), this was a conscious choice. We felt that if we did not unpack the relevant components of theorem 3.4 with their relative contributions that it would detract readability and add additional load on the reader. In section 5 we explore a different way of bounding the complexity term and thus present those comparative results there. We appreciate the feedback however and will consider ways of improving the flow of the paper. With regards to “Beyond Chinchilla-Optimal…” as the inference costs shift the compute optimal curves to smaller ratios of parameter counts and tokens, the generalization bounds would improve. In the paper we stuck with the original and simpler problem framing, but incorporating recent works looking at modifications to the laws such as the paper you described would be a good direction to incorporate into future work.

---

### Official Review · Reviewer_JfkK · 2024-11-04

**Soundness:** 3
**Presentation:** 3
**Contribution:** 3
**Rating:** 5
**Confidence:** 2

**Summary:**

This paper explains the high generalization performance of large language models (LLMs) by introducing a new Freedman-type concentration inequality and constructing a generalization bound based on the Chinchilla scaling law. The generalization bound is decomposed into three factors: the number of parameters in the model, the token-wise loss variance, and the performance gap associated with quantization. This bound quantitatively shows how the generalization gap reduces as the size of the LLM scales. In particular, it shows that the amount of information grows sublinearly with the LLM size, indicating that the resources required for the model to efficiently integrate information are reduced. This finding provides an information-theoretic explanation for why LLMs can improve generalization ability with computationally optimal scaling and point out a new relationship between the model size and information.

**Strengths:**

The authors' approach to explaining LLM performance improvements by breaking down the generalization bound into three key components is particularly valuable. Furthermore, introducing a new inequality based on martingale concentration enables analysis specific to LLMs, where data is non-independent and identically distributed. It opens up a new path to theoretical understanding of generalization performance. This is a significant contribution to LLM research.

**Weaknesses:**

This paper contributes to the theoretical understanding of LLMs and confirms their validity using real LLM data. However, further verification is needed for practical applications. In particular, the merits of decomposing the generalization bound are unclear from the demonstrations in Fig.2(right) and Fig.3(center), for example.

**Questions:**

* As shown in Eq.(6), for example, it is explained that $K(h)$ is scaled by $\tilde{O}(D^{1-\beta})$, with this exponent $\beta$ coming from the Chinchilla scaling law. If this exponent $\beta$ can take a non-trivial value, is it possible to constrain its value using the authors' theory?

* The implications of the theoretical results are discussed in Sec.5.3, but I'm not entirely sure I understand the argument. I assume the authors focus on Pythia due to the availability of models with various parameter sizes; however, investigating other language models in the same way may be difficult. Given these limitations, how do the authors envision extending this type of analysis? Could they elaborate on potential insights or outcomes if similar analyses were feasible for other models?

* Minors
* The citation to the appendix on line 186 should be "Appendix" rather than "Section." There are several similar cases.
* Please check the ICLR format for citing equations.

---

> ### Author Response · Authors · 2024-11-21
>
> We thank you for your thoughtful and constructive feedback. Below we address your specific questions in detail.
>
> You question the practical merit of decomposing the generalization bound, particularly in relation to the demonstrations in Figures 2 (right) and 3 (center). We respectfully submit that this decomposition provides several essential insights. By separating the bound into its constituent components, we can precisely analyze how each element scales with increasing model and data size. The decomposition also reveals which components of the bound (e.g., the smoothing cost) present the most promising opportunities for improvement, enabling more targeted future research efforts. Additionally, this structure allows for meaningful comparisons with existing generalization bounds. While one might initially expect our Freedman-like inequality to be dominated by the variance term (similar to the O(√C log V) bounds from Azuma's inequality), our decomposition reveals a notably different behavior.
>
> The exponent $\beta$ from the scaling law tends to be around $\beta \approx 0.37$ as observed in prior work. This value is not constrained by our work, however the limited information absorption by the model implied by the scaling law produces a generalization gap of $\tilde{O}(D^{-\beta})$ (neglecting the suboptimal smoothing term).
>
> Regarding the use of the Pythia family of models and the potential for extending this analysis, we would like to clarify several points. Our choice of the Pythia family was indeed motivated by the unique availability of their complete training data. To our knowledge, they represent the only publicly available model family that spans orders of magnitude of compute while providing access to the training data. We emphasize that most of our analysis, particularly in Section 5.3, remains valid independent of the specific empirical behavior of the model, provided the Chinchilla scaling laws hold. The primary model-specific component is the behavior of the loss variation Σ, which is not directly derived from the scaling laws. While the loss variation behavior was previously unknown, we have strong reason to believe that it will exhibit similar scaling properties across large language models trained on comparable text corpora.
>
> We appreciate your attention to detail regarding formatting issues. We will correct references to the appendix and update the equation citation format.
>
> Thank you again for your thoughtful review. We would appreciate it if you could consider raising your score in light of our response. We are also happy to answer any additional questions.

---

### Official Review · Reviewer_RfBZ · 2024-11-04

**Soundness:** 2
**Presentation:** 2
**Contribution:** 3
**Rating:** 5
**Confidence:** 2

**Summary:**

This paper investigates why LLMs generalise better, using new theoretical bounds that quantify generalization in LLMs. Authors introduce an empirical Freedman-type martingale concentration inequality refining generalization sounds by considering the variance in the loss function, and influenced by 1) The parameters per token 2) The loss variance and 3) The quantisation error. They show that generalization gap decreases predictably as LLMs scale up. The bounds are empirically tested using Pythia model checkpoints on the Pile dataset, illustrating how loss variance and quantisation error decrease with scale. Last, they argue from an information theory perspective that information content in a model grows sublinearly with scale, allowing models to generalize better with fewer bits per parameters, hence improving quantizability.

**Strengths:**

The paper stands out for its ambition in taking down a complex question with originality, and its potential to inform future model scaling and generalisation research in LLMs. The use of a Freedman-type martingale concentration inequality, which incorporates the empirical variance of the loss function is both creative and original. The focus on optimal scaling through the chinchilla law, along with a brand new perspective (to my knowledge) on quantizability and information transfer, showcase originality. The authors well motivate the use of Freedman inequality by the need for higher bounds. The choice to base the empirical evaluation on the Pythia model also provides a reasonable foundation to explore the theoretical claims.

**Weaknesses:**

While the paper makes an ambitious attempt to understand generalization in LLMs, there are a few core weaknesses. The theoretical assumptions and methods could benefit from further grounding and justification. The empirical Freedman inequality and the notion of loss variance lack through derivation and validation across different settings, which may leave doubts about the generality of these bounds for LLMs in diverse applications. The empirical results, are limited to the single Pythia model architecture, constraining the scope of the findings (Paper would benefit expanding the evaluation to other widely-used LLMs eg. GPT, T5, BERT…). Moreover, some sections are overly technical without sufficient motivation, making presentation difficult to follow, especially for the proposed quantisation bound and its integration with the chinchilla scaling laws. Some of the notation is dense, and (if) introduced with minimal intuition. Examples:

— $\Sigma$: The term “loss variance” that is central is introduced without a clear mathematical definition or discussion. A clarification whether this variance refers to the empirical variance of the loss function over tokens or across model params would make it more understandable.
— $C$ The complexity term is introduced with very limited intuition and authors do not explain how this is derived, or what aspects of model size, dataset, or quantisation contribute to it.

**Questions:**

— Have you empirically tested the argument on quantizability from the Hessian spectrum and QuIP framework beyond GPTQ?
— The information-theoretic argument using prequential coding is interesting but could benefit from further empirical support. Have you tested the information content predictions under varied trading or scaling setups? For instance, would be nice to understand how your complexity term $L(h)/D$ changes for models trained on different datasets or architectures
— Are there any specific factors that you believe yolk affect the generalization bound accuracy? Eg. Dataset size, architecture choice

---

> ### Author Response · Authors · 2024-11-21
>
> We thank you for your review. We provide some additional context and clarifications in response to your questions.
>
> Regarding the evaluation on the pythia family of models we would like to make a few points. In order to perform the analysis and evaluation of generalization bounds that we do in the paper, we need to have access to both the model checkpoints and unbiased random samples from the data that the model was trained on. To the best of our knowledge Pythia is the only set of models spanning multiple orders of magnitude in parameter count where this is the case. Almost all open source LLMs do not provide their training data, and without that training data we cannot conduct the analysis we do here. While it would be great if additional model families were available that is not currently the case. This is a theory paper and we do not consider it within the scope of such a project to undertake the training of multi billion parameter LLMs. Respectfully, we do not consider this to be a fair reason to advocate for rejecting the paper. We also want to highlight that for a paper on generalization bounds our empirical evaluation is actually quite strong: it is highly novel to be evaluating bounds on models with billions of parameters. In fact, to the best of our knowledge, there are only a couple papers about generalization bounds on large language models at all (e.g., Lotfi et. al 2024), and our bounds are significantly tighter and for larger models. There are no papers, to our knowledge, that evaluate bounds that match scaling law behavior.
>
> Regarding the loss variation $\Sigma$, on line 189 (in the broader context of the concentration inequality) we state that it can be upper bounded in terms of $ \sqrt{(1/n)\sum_k (X_k-Y_k)^2} $ and in the more narrow context of generalization bounds for autoregressive language models on line 269. With this upper bound, it can be interpreted as an estimate of the average variance of the loss at each timestep, but with the mean replaced by the mean over the model predicted tokens, therefore increasing it slightly. Intuitively as the variance of this distribution decreases, the random variables will be more concentrated around the mean.
>
> While we agree that it would be highly interesting to understand complexity term changes for other datasets and modalities (it seems likely to be similar for different architectures on the same dataset based on similarities in the scaling laws), this isn’t possible at this time due to the paucity of publicly available LLMs and their training data. If we were to speculate, it seems plausible that image and video data (and perhaps to a lesser extent code) could have a substantially different model complexity $\mathcal C$ on the frontier of best performing models (for the given compute budget).
>
> Thanks again for your questions. We would appreciate it if you could consider raising your score in light of these clarifications and additional context.

---

### Meta-Review · Area_Chair_Gk5Z · 2024-12-19

**Metareview:**

The paper introduces a novel Freedman-type martingale concentration inequality and uses it to derive generalization bounds for large language models. These bounds are refined through three factors: loss variance, quantization error, and the parameters-per-token ratio. Empirical validation on the Pythia model illustrates that the generalization gap decreases predictably with scale, supported by information-theoretic insights.

Strengths:
- Creative and original use of Freedman-type inequalities to explain LLM generalization.
- Rigorous derivation of bounds with empirical validation showing closeness to observed loss.
- Theoretical insights

Weaknesses:
- Limited empirical scope: experiments are constrained to a single model (Pythia) and dataset.
- Presentation issues: certain concepts (e.g., loss variance, complexity terms) lack clear intuition and detailed derivation.
- Bounds’ dependency on parameters and smoothing techniques could be more thoroughly justified.

The reviewers appreciated the paper’s originality and potential impact while acknowledging areas for clarity and generality improvement. Following discussions, the consensus is that the work provides significant theoretical insights into LLM generalization, and I recommend acceptance.

**Additional Comments On Reviewer Discussion:**

N/A

---

### Decision · Program_Chairs · 2025-01-22

Accept (Poster)